# CattleNet-XAI: An explainable CNN framework for efficient cattle weight estimation

**Md Junayed Hossain, Jannatul Ferdaus, Ashraful Islam**⬥**\*, M. Ashraful Amin**⬥

Center for Computational & Data Sciences, Independent University, Bangladesh, Dhaka, Bangladesh

\* ashraful@iub.edu.bd

## Abstract

Accurate estimation of cattle weight is essential for effective farm management, health assessment, and productivity optimization. Traditional manual methods for weight estimation, however, are labor-intensive, time-consuming, and prone to inaccuracies. Recent advances in computer vision have facilitated the automation of weight prediction from image data. However, traditional regression models, such as Random Forest and Linear Regression, face challenges in capturing the complex, nonlinear relationships within image data, leading to less accurate predictions. To address these issues, we introduce CattleNet-XAI, a framework designed for both efficiency and explainability, which utilizes a custom Convolutional Neural Network (CNN). For the CNN-based approach, we incorporated advanced image preprocessing techniques, including normalization and histogram equalization, to enhance the input data quality. We compared its performance with other CNN models, the pretrained EfficientNetB3 model, and traditional machine learning methods like Random Forest and Linear Regression. For the traditional methods, we first leveraged the YOLOv5 algorithm for precise feature extraction from the cattle images. All the models were trained and evaluated on a dataset of cattle images and associated weight data, with performance measured by Mean Absolute Error (MAE), Mean Squared Error (MSE), and Root Mean Squared Error (RMSE). Our results show that one variation of our custom CNN (3Conv3Dense) model significantly surpasses other conventional models, achieving a low MAE of 18.02 kg and an RMSE of 19.85 kg, which demonstrates superior accuracy. We also present LIME visualization and error case analysis to provide insights into the decision-making process of the model. This study emphasizes the capability of deep learning, especially CNN, in automating and enhancing the precision of livestock weight estimation, offering a modern and effective approach to cattle management.

## 1 Introduction

Livestock is essential to the agricultural economy in Bangladesh, serving various functions in the lives of countless farmers. With an estimated 23.78 million cattle,

**Data availability statement:** The data underlying the results presented in the study are available from the GitHub Repository - CID: Cow Images dataset for regression and classification (https://github.com/bhuiyanmobasshir94/CID).

**Funding:** The author(s) received no specific funding for this work.

**Competing interests:** The authors have declared that no competing interests exist.

livestock is essential for food security, nutrition, income development, savings, draft power, manure production, and transportation, in addition to fulfilling important social and cultural roles [1]. It constitutes a vital source of "cash income" for many rural farmers, as it can be readily acquired through sale or barter, thus providing financial flexibility during challenging times [2]. Cattle fattening has emerged as a vital sector for small-scale farmers, serving as an effective strategy for reducing poverty. This activity has become significant in both rural and urban populations, as it offers employment opportunities and a source of income [3]. The growing significance of cattle fattening highlights the need for precise and effective techniques to manage and improve cattle farming, especially weight estimation, which is crucial for market value and economic sustainability.

Given the critical importance of cattle in maintaining the economic stability of rural communities, accurate weight measurement is essential for effective livestock management [4]. This encompasses the allocation of feed, observation of cattle health, and assessment of appropriate market prices. Historically, several systems have been utilized to measure the live weight of cattle. Like Weighbridge [5] serving as the principal standard, alongside formulaic methods such as Agarwal's and Shaeffer's formulas [6], which calculate weight based on girth and length measurements. The Rondo tape and Weigh tape techniques [7] involved the objective measurement of certain anatomical locations, with results either recorded promptly or calculated using predetermined charts. Moreover, the Calculator technique utilized heart girth measures to assess body weight [8]. However, all of these conventional methods necessitate considerable effort, are time-consuming, and are frequently susceptible to human error, suggesting that the weight estimations may not always be accurate.

Recent research has increasingly turned to Computer Vision (CV) [9], Machine Learning (ML) [10], and particularly Convolutional Neural Networks (CNNs) [11], to overcome the limitations of manual weight estimation. However, its application in the agricultural sector, particularly for predicting livestock weight, is still in the early stages. Current approaches reported in the literature can be broadly categorized into two groups: those based on 3D imaging and those utilizing conventional 2D images. Methods employing 3D vision, such as those using depth maps or stereo vision, have shown promise but often rely on expensive, specialized equipment and complex preprocessing pipelines, making them less accessible for many farmers. On the other hand, approaches using 2D images are more cost-effective but face the challenge of accurately inferring a 3D characteristic (weight) from a 2D plane. While CNNs have proven effective for extracting relevant features from 2D images, many existing models function as "black boxes", lacking the transparency needed for user trust and adoption in practical farming scenarios. This highlights a critical need for a solution that is not only accurate and cost-effective but also interpretable, allowing users to understand the basis of the weight prediction.

This paper presents a comprehensive study of two distinct approaches for predicting cattle weight using image-based data, aiming to enhance both predictive accuracy and model robustness. The first approach leverages CNN models, incorporating various architectures, including EfficientNetB3 [12] for weight estimation. To augment the model's efficacy, various image processing approaches, such as

Normalization and Histogram Equalization are employed to boost the quality of the input data. Among the various CNNs model architectures the 3Conv3DenseNet Fig 6(a), attained the best Mean Absolute Error (MAE) [13] of 18.02 kg, and a Mean Absolute Percentage Error (MAPE) [14] of approximately 6.22%, which demonstrating the effectiveness of CNNs in estimating cattle weight. The second technique focuses on the extraction and selection of features from cattle images. Utilizing the object detection capabilities of the You Only Look Once (YOLOv5) [15] algorithm, pertinent features are collected from the images and meticulously refined through a stringent feature selection procedure [16] to ascertain the most predictive features. The chosen features are subsequently employed in regression models, namely Linear Regression (LR) [17] and Random Forest Regression (RFR) [18], to estimate the weight of the cattle. The LR model yielded an MAE of 25.99 kg, but the RFR model attained an MAE of 23.67 kg, illustrating the efficacy of integrating feature engineering with conventional regression methods. Here we also applied Local Interpretable Model-agnostic Explanations (LIME) [19], to understand the model's weight predictions and locally faithful explanations. Which evaluates the amount of confidence in human users of the Artificial Intelligence (AI) system [20]. And this experiment utilized the Cow Image Dataset (CID) [21], comprising around 17,899 images of cattle along with several additional attributes.

The contribution of this paper can be summarised as follows:

- **CNNs-Based Weight Estimation Model:** This research introduces a custom CNNs model (3Conv3Dense Fig 6(a)), that incorporates advanced image preprocessing techniques, including normalization and histogram equalization, resulting in a MAE of 18.02 kg. This contribution highlights the efficacy of CNNs for cattle weight estimation.
- **Integration of Object Detection and Regression Techniques:** By leveraging the YOLOv5 model for precise feature extraction from cattle images, followed by the application of LR and RFR. This highlights the efficacy of combining feature engineering with traditional regression models to enhance prediction accuracy.
- **Integration of Explainable AI (LIME) in Livestock Weight Prediction:** This study employs LIME to investigate how the models predict cattle's weight. Applying LIME, we not only predict weights, but also reveal which body regions (rib cage, abdomen, hindquarters) drive predictions. This novel integration addresses the critical issue of trust and transparency in AI adoption within agriculture.
- **Utilization of the Cow Image Dataset (CID):** The research leverages the Cow Image Dataset (CID), containing a substantial collection of approximately 17,899 cattle images across 8 different breeds. The large size of this dataset ensures that the models are well trained and stable, providing a comprehensive validation framework that significantly enhances the robustness of the research findings.
- **Error Case Analysis for Practical Relevance:** Beyond standard performance metrics, we present a detailed residual and error analysis to identify the conditions under which the models underperform. This contribution, often overlooked in prior work, offers practical insights for improving future systems deployed in farms.

This paper is structured in the following manner: Sect 2 provides a summary of the latest methodologies relevant to our research. Sect 3 defines the problem statement and proposes a potential solution. Sect 4 outlines the proposed architecture. Sect 5 summarises the results, while Sect 6 concludes the study and discusses potential future research.

## 2 Literature review

Precise livestock weight estimation is integral to optimizing agricultural operations, influencing feeding strategies, health monitoring, and market valuation. Traditional weight measurement techniques, although widely used, are often laborious and susceptible to inaccuracies. Recent developments in computer vision and machine learning have introduced more efficient, non-invasive methods that offer enhanced accuracy and automation in predicting livestock weight, which making a significant advancement in agricultural technology.

Ruchay et al. [22] introduce a novel approach to predicting cattle live weight using deep learning-based image regression with RGB and depth maps. Their method features efficient preprocessing techniques and 3D augmentation methods, resulting in an automated, non-contact weight estimation system that achieves a prediction accuracy of 91.6%. Building on this, Beibei et al. [23] employ semantic segmentation and BP neural networks for weight prediction. Utilizing the ResNet-101-D architecture with Squeeze-and-Excitation attention, their model excels in extracting body size parameters, achieving a MAE of 13.11 pounds and an RMSE of 22.73 pounds.

Similarly, X. Song et al. [24] employ 3D vision technology to automate dairy cattle weight prediction by extracting key morphological traits such as hip height, hip width, and rump length. Combined with factors such as Days in Milk (DIM), age, and parity, their multiple linear regression model achieved an RMSE of 41.2 kg and a MAPE of 5.2%. While the study utilized 3D vision, Chang et al. [25] addressed a 3D segmentation-based method for predicting Korean cattle weight. By employing PointNet for 3D segmentation and random forest regression, they achieved a MAE of 25.2 kg and a MAPE of 5.85%, demonstrating the effectiveness of this non-invasive approach.

In other work, Alexey et al. [26] propose a method for predicting farm animal weight using stereo vision and deep learning-based semantic segmentation. They captured images from side and back angles, processed with the Deeplab v3+ model to extract size and distance features. Among the three trained ANN models, the model using both side and back images achieved the best accuracy, with a margin of error around ±20 kg.

Rather than relying on 3D vision or segmentation, Mikel et al. [27] applied deep learning techniques to predict beef cattle body weight using 2D images. They addressed the challenge of information loss when converting 3D object shapes into 2D data by testing CNN and recurrent attention models. The best-performing model, EfficientNet-B1, achieved a MAE of 21.64 kg, significantly reducing the error from traditional linear regression models (38.46 kg).

Similar to this, Roel et al. [28] introduced a novel deep learning-based model for predicting the body mass of heifers using 2D images, focusing on both top and side views. By leveraging the Mask-RCNN segmentation algorithm to remove the background and a CNN-based model for prediction, the top-view model achieved superior accuracy with an R² of 0.96 and an RMSE of 20 kg, while the side-view model reached an $R^2$ of 0.91 and an RMSE of 27 kg.

In another Ye Bi et al. [29], proposed a model to predict the longitudinal body weight of dairy cows using depth video data and evaluated three segmentation methods: single-thresholding, adaptive-thresholding, and Mask R-CNN. Depth images were collected using an Intel RealSense D435 camera, and biometric features such as volume, height, width, and length were derived. Regression models, including linear mixed models (LMM), were applied to predict body weight. Mask R-CNN combined with LMM achieved the best prediction performance with an R² of 0.98 and a mean absolute percentage error (MAPE) of 2.03%.

On the other hand, Oleksandr et al. [30] proposed a method for predicting cattle weight using machine learning algorithms by capturing images from multiple angles. The system uses the Mask-RCNN algorithm and stereopsis methods to estimate body dimensions such as withers height, hip height, body length, and hip width. These dimensions are then fed into a neural network (MLP) to predict the live weight of cattle.

Shifting from image-based methods, Ruchay et al. [31] employed machine learning algorithms to predict the body weight of Hereford Cattle based on morphological traits such as withers height, hip height, and chest girth. Among the tested models, the Random Forest Regressor achieved the best performance with an $R^2$ of 0.644, an RMSE of 37.259 kg, and a MAE of 24.965 kg on the testing dataset. This highlights the potential of integrating both image-based and morphological data for weight prediction.

Similar to this shifting from cattle to pig, Gaganpreet et al. [32], used a smartphone app, On 3D CameraMeasure, to predict the weight of Ghoongroo pigs from images. A formula based on body length and heart girth height showed excellent accuracy with an $R^2$ of 0.98. This approach is suitable for both organized farms and rural areas with minimal resources.

When the livestock is sheep, William et al. [33], applied six ML algorithms, including RF and eXtreme Gradient Boosting (XGBoost), to predict body weight in Peruvian Corriedale sheep using 14 body measurements. After feature selection,

RF outperformed other models with the highest $R^2$ of 0.92 and lowest errors (MAE = 2.83, RMSE = 3.75) on the testing dataset. This method offers an accurate, efficient, and low-cost alternative for weight estimation, benefiting researchers and smallholders.

While livestock weight estimation using 2D and 3D imaging has improved, challenges remain in achieving accuracy, efficiency, and adaptability. Many methods rely on complex 3D imaging systems that need expensive equipment and lengthy preprocessing. Some models are hard to understand, making them less useful for farmers and agricultural workers. Additionally, the datasets used are often not diverse enough, which limits the ability of these models to work well across different environments and cattle breeds. To overcome these challenges, we developed CattleNet-XAI, a simple and efficient framework using 2D cattle images. This work provides a practical, cost-effective, and reliable solution for precise cattle weight estimation.

## 3 Proposed architecture

The proposed methodology for cattle weight prediction involves two distinct approaches: the Backpropagation Method and the Feature Extraction Method. These methods are applied in sequence to analyze a dataset containing cattle images, and both aim to predict the cattle's weight efficiently by leveraging traditional regression model and advanced CNN techniques.

The Backpropagation Method focuses on direct image processing [34]. Images are first converted into arrays, allowing for numerical manipulation. Next, normalization is applied to standardize pixel values then applied histogram equalization to enhance the image contrast. The preprocessed images are then used in a Backpropagation algorithm, which is a type of neural network, to predict the cattle's weight.

In the Feature Extraction Method, cattle images are first detected and extracted. Feature extraction is then performed to isolate important characteristics from the cattle images, such as texture, size, and shape. Afterward, a feature selection process is employed to choose the most relevant features for weight prediction. These selected features are then fed into a regression algorithm to predict the cattle's weight.

Finally, both approaches are analyzed by comparing their results, and the best-performing method is selected based on MAE, MAPE and RMSE. This comparison provides insights into which technique offers the most reliable predictions, helping in the selection of the most appropriate model for cattle weight prediction tasks. We then used LIME to provide a visual explanation of how our best model predicts cattle weights based on their images. Each of them is described in brief in the sections below, and the proposed architecture is visually illustrated in Fig 1.

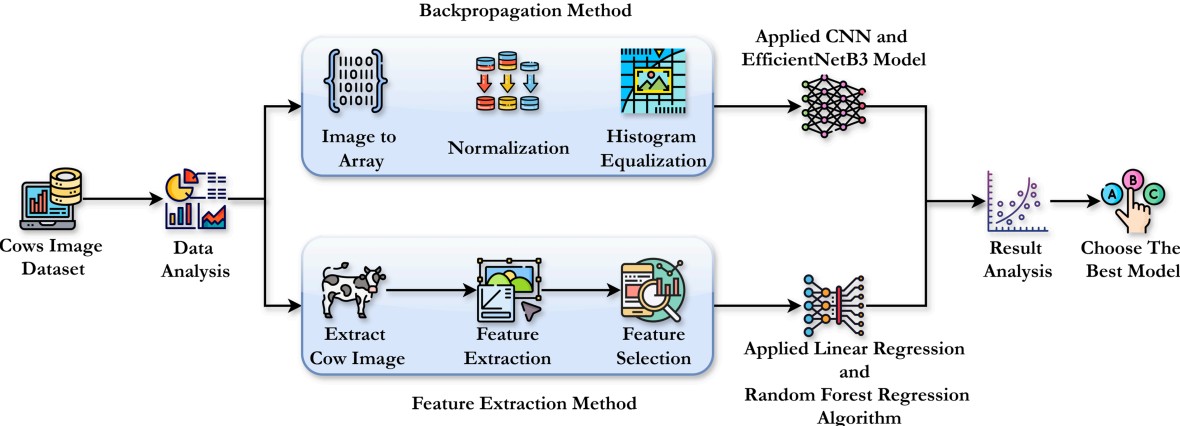

**Fig 1**. Proposed architecture of our research.

### 3.1 Data description

The exponential growth in cattle livestock market has created significant challenges for managing production and procurement. To address these challenges, we aim to leverage emerging technologies and advanced CV techniques. For our research, we utilized an existing dataset known as the Cow Image Dataset (CID) [21]. This dataset has 17,899 labeled images of cattle, sourced from both taken images of original cattle and YouTube extractions. We have only acquired 2052 original images of a total of 513 cattle. Images are provided, each annotated with essential data like colour, breed, feed type, age, teeth count, height, weight, price, and size. In Fig 2, we present a sample from the dataset. This dataset is versatile, supporting both classification and regression tasks, and is a valuable resource for researchers working on cattle livestock analysis.

The scatter plot in Fig 3(a), shows the relationship between the height (in inches) and weight (in kilograms) of cattle. As indicated by the clustering of points, there is a positive correlation between height and weight, meaning that as the height of a cattle increases, its weight generally increases as well. The data points, however, show some variability, especially at higher heights, where the weight can vary significantly. This suggests that while height is a good predictor of weight, other factors may also influence the weight of the cattle. The plot provides a visual representation of this trend, helping in analyzing the relationship between these two important variables.

The pie chart in Fig 3(b), displays the distribution of different cattle breeds. The majority of the dataset consists of LOCAL breed cattle, making up 68.8% of the total. Other significant breeds include SAHIWAL (11.5%) and SINDHI (7.4%). Smaller breed representations include HOSTINE CROSS (6.2%), RED CHITTAGONG (4.1%), and PABNA BREED (1.2%). The rarest breeds in the dataset are BRAHMA and MIR KADIM, each accounting for only 0.4% of the

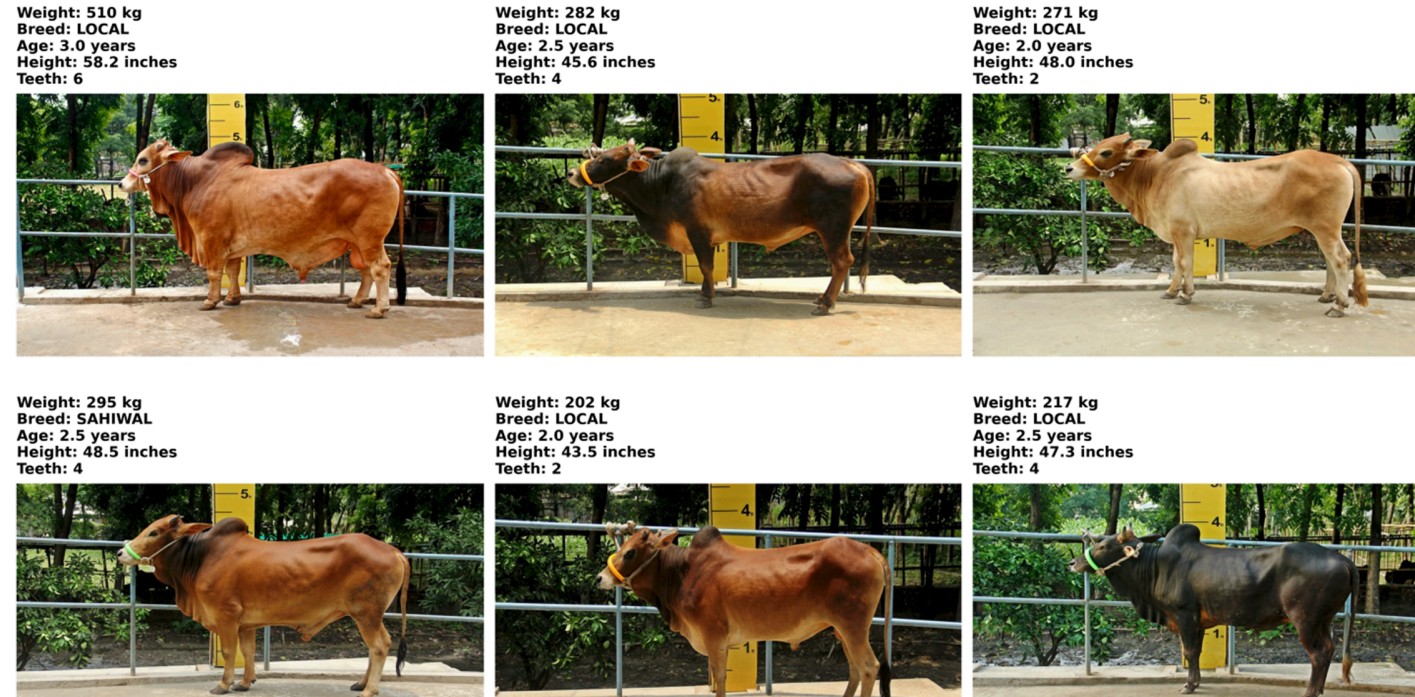

**Fig 2**. A summary of the dataset containing information on each cattle's weight, breed, and height.

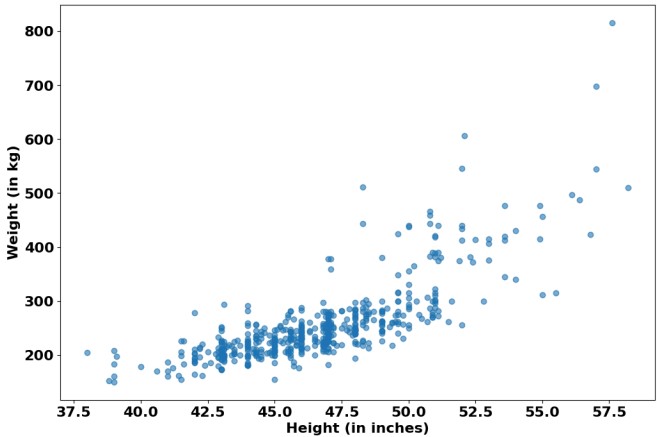

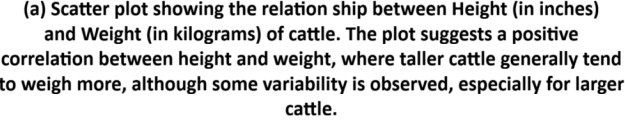

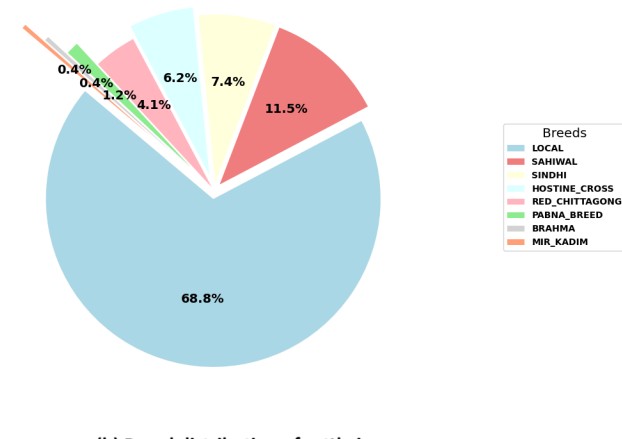

(a) Scatter plot showing the relation ship between Height (in inches) and Weight (in kilograms) of cattle. The plot suggests a positive correlation between height and weight, where taller cattle generally tend to weigh more, although some variability is observed, especially for larger cattle.

(b) Breed distribution of cattle in the dataset, showing that the LOCAL breed constitutes the majority.

**Fig 3. Height-Weight correlation and breed distribution of cattle in the dataset.**

total. The chart highlights the dominance of the LOCAL breed in the dataset. Despite this imbalance in breed distribution, oversampling of the minority classes is not required in this case. The predictive model primarily focuses on body features such as the rib cage, abdomen, and overall body shape, which are mostly consistent across breeds included in the dataset. The variations among breeds, largely limited to color, head shape, and horn structure that do not significantly impact the model's weight prediction accuracy.

## 3.2 Image processing

The proposed architecture in this paper introduces two key methodologies for predicting cattle's weight: a CNNs based approach and a Feature Extraction combined with Regression approach. Below is a comprehensive outline of the data preprocessing steps specific to each method, aligned with the framework presented in this research.

**Image processing for CNN-based weight estimation:** In the process of predicting cattle weight using CNN and EfficientNetB3, the order of processing steps is crucial for ensuring the model's performance and accuracy. The processing sequence applied in this study consists of Image to Array Conversion, Normalization, and Histogram Equalization, each step designed to prepare the data in a structured manner that enhances the model's ability to learn effectively.

**3.2.1 Image to array conversion.** The first step is converting the cattle images into numerical arrays. This step is fundamental because it transforms the raw image data into an array so that CNNs can interpret. Each pixel in the image is represented as an intensity value, forming an array that can be fed into the CNNs for further processing. Without this step, the CNNs would not be able to process the images computationally [35].

For an image with dimensions $h \times w \times c$ (height, width, and color channels), the image is transformed into a matrix $I$, which allows the CNN to apply convolutional filters to extract important features.

$$I = \{p_{i,j,k} \mid 0 \le i < h, 0 \le j < w, 0 \le k < c\} \tag{1}$$

Where $p_{i,j,k}$ represents the pixel intensity at position $(i,j)$ for channel $k$. This numerical representation enables the CNNs to perform its operations on the image and learn the relevant features for weight prediction.

**3.2.2 Normalization.** Normalization is applied to standardize the pixel values to a common range, typically between 0 and 1 [36]. This step is done before histogram equalization because normalization ensures that the intensity values, after contrast adjustments, are scaled to a consistent range, which accelerates the learning process during CNNs training [37]. Normalization also prevents large variations in pixel values from distorting the learning process and ensures that the CNNs model operates effectively across all input images.

The normalization formula is as follows:

$$X_{norm} = \frac{X - \min(X)}{\max(X) - \min(X)} \tag{2}$$

Where:

- $X_{norm}$ is the normalized pixel value,
- $X$ is the original pixel value,
- $\min(X)$ and $\max(X)$ are the minimum and maximum pixel values in the image.

**3.2.3 Histogram equalization.** Once the images are normalized, histogram equalization is applied. This step improves the contrast of the image by redistributing the intensity values, making important features like the cattle's body shape and size more distinguishable [38]. And this histogram equalization step is applied after the normalization because the intensity values are now accessible in a numerical format, which allows for contrast adjustments to be made systematically.

The transformation for histogram equalization is given by:

$$s = T(r) = (L - 1) \int_0^r p_r(w)dw \tag{3}$$

Where:

- $r$ is the original intensity,
- $s$ is the new intensity after equalization,
- $L$ is the total number of intensity levels,
- $p_r(w)$ is the probability density function of the pixel intensities.

Applying this step after normalization is essential, as it directly modifies the pixel values in a way that spreads out the intensity range, making features in the image more prominent for the CNNs to detect.

The sequence of these processing steps is crucial because each step depends on the output of the previous one. First, image-to-array conversion must be performed since the raw image data needs to be transformed into a numerical format that the CNNs can process. This transformation enables the CNNs to perform operations on the image, such as convolution, to extract features. Next, normalization is performed to scale the pixel values to a consistent range, typically between 0 and 1. This step is important before histogram equalization because it stabilizes intensity values, which helps speed up CNN training and ensures the model converges well. Finally, histogram equalization is applied, which adjusts the contrast of the image by redistributing the pixel intensity values. This step is most effective when applied to the original intensity data in array form, ensuring that important features, such as body contours, are more distinguishable. Here in Fig 4, show each of the processing steps. After processing the images, our proposed custom CNNs algorithm trained to predict the weights.

**Image processing for feature extraction and regression-based weight estimation:** This section outlines the process of extracting cattle images, performing feature extraction using YOLOv5, and selecting important features for weight

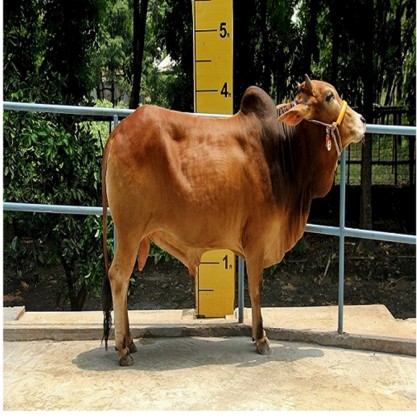
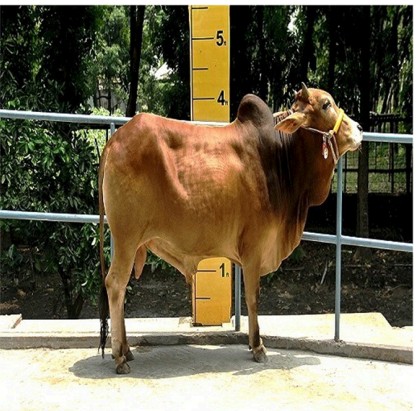

(a) Original Image.

(b) Image after Array Conversion.

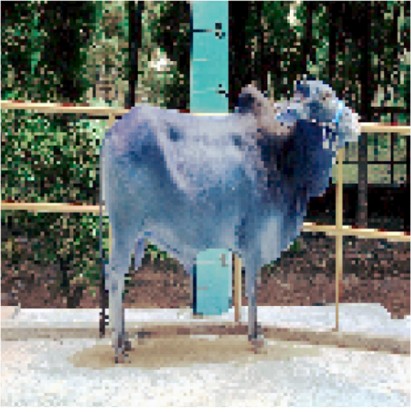

(c) Image after Normalization.

(d) Image after Histogram Equalization.

**Fig 4**. Image processing steps applied in our research.

estimation. The extracted features are processed through machine learning models like Linear Regression and Random Forest Regression. Key techniques, including recursive feature elimination, are employed to enhance the predictive accuracy of the regression models for cattle weight estimation. The details are described below.

**3.2.1 Cattle image extraction from original image dataset.** The first step in this process is to extract the cattle images from the original dataset, which consists of 17,899 images labeled with various attributes, such as weight, breed, and height. This step involves identifying the region of interest, primarily focusing on the cattle's full body, which will later be used for feature extraction and weight estimation.

We applied YOLOv5 to detect and isolate the cattle from the background. YOLOv5's object detection algorithm operates by generating bounding boxes around the object of interest (in our case, the cattle), which is represented as:

$$B = \{(x_{min}, y_{min}), (x_{max}, y_{max})\} \tag{4}$$

where $(x_{min}, y_{min})$ and $(x_{max}, y_{max})$ are the coordinates of the top-left and bottom-right corners of the bounding box, respectively. This ensures that only the cattle's body is extracted while excluding unnecessary background elements.

The extracted image is then resized to a uniform dimension $W \times H$ (width by height), where each cattle image conforms to the same input size required for the feature extraction process that follows. Here in Fig 5 we can see the extracted image from the original image.

**3.2.2 Feature extraction using YOLOv5.** Once the cattle images are isolated, feature extraction is performed to capture the essential characteristics that correlate with the cattle weight. We utilize the YOLOv5 model to extract relevant features [39], such as the cattle's body size, shape, and texture. The bounding box coordinates generated in the previous step, $B = \{(x_{min}, y_{min}), (x_{max}, y_{max})\}$, are used to define the region of interest. Although there are recent versions of YOLO with improved features and capabilities, we chose to employ YOLOv5 because of its computational effectiveness and fit for the particular needs of our study. Our goal is to extract features from cattle images. Without imposing an unnecessary computational burden, YOLOv5 is perfect for extracting crucial properties like body dimension, texture, and shape since it offers a fair trade-off between performance and resource utilization. This version meets the objectives of our study and is robust enough for us to obtain accurate and dependable feature extraction.

The features extracted can be represented as a vector $F$, where each element in the vector corresponds to a specific attribute of the cattle's image:

$$F = \{f_1, f_2, f_3, \dots, f_n\} \tag{5}$$

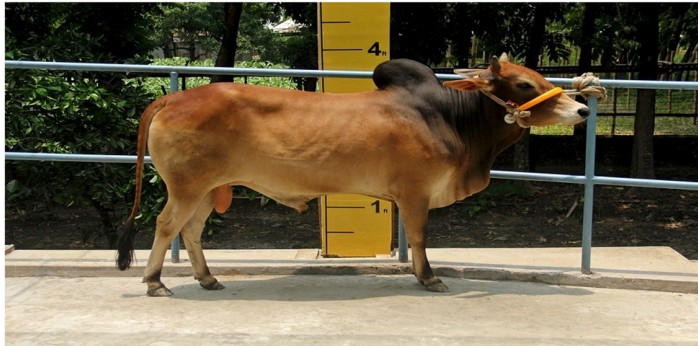

(a) This is the original cattle image from our dataset, with a resolution of 1200x700 pixels

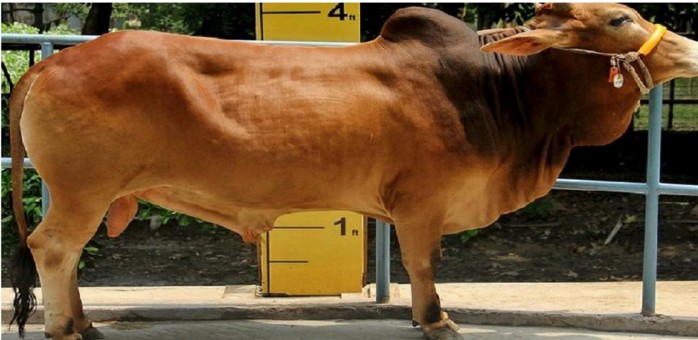

(b) This is the cattle's image after being extracted using YOLOv5, with a resolution of 496x349 pixels.

**Fig 5**. Here is the extracted cattle's image from the original image using YOLOv5 algorithm, which accurately detects and isolates the cattle for further processing.

Here, $f_i$ denotes individual features such as body length, height, width, and pixel-based dimensions. These features are calculated from the bounding box dimensions, for instance:

$$\text{Height} = y_{max} - y_{min}, \quad \text{Width} = x_{max} - x_{min} \tag{6}$$

Other extracted features include the aspect ratio:

$$\frac{\text{Height}}{\text{Width}} \tag{7}$$

These pixel intensity variations contribute to predicting the cattle's weight. The YOLOv5 model applies its convolutional layers to the region of interest, capturing spatial hierarchies of features that are crucial for understanding the cattle's physical structure.

**3.2.3 Important feature selection.** After extracting a comprehensive set of features, it is necessary to select the most relevant ones that contribute effectively to weight estimation. To achieve this, we use Recursive Feature Elimination (RFE) [40], a feature selection technique that iteratively removes the least important features based on model performance. The objective is to retain only the most significant features that have the highest correlation with the cattle's weight.

Let $X$ represent the matrix of features extracted from the cattle images, where each row corresponds to an image and each column corresponds to a feature:

$$X = \begin{pmatrix} f_{11} & f_{12} & \dots & f_{1n} \\ f_{21} & f_{22} & \dots & f_{2n} \\ \vdots & \vdots & & \vdots \\ f_{m1} & f_{m2} & \dots & f_{mn} \end{pmatrix} \tag{8}$$

where $m$ is the number of images, and $n$ is the number of features.

Through RFE, the less significant features are iteratively eliminated, and only top 100,000 features from the original feature set are selected. This reduced feature space was then used to train regression models such as LR and RFR for predicting cattle weight. most informative features are retained. While the selected features from RFE are abstract and do not directly map to specific anatomical regions, we complemented our analysis using LIME to understand the spatial importance of image regions. Visualizations generated by LIME across multiple test samples consistently highlighted the rib cage, abdomen, and hindquarters as the most influential areas for weight prediction. The selected feature set $F'$ can be represented as:

$$F' = \{f'_1, f'_2, \dots, f'_k\}, \quad k \leq n \tag{9}$$

This refined feature set $F'$ is then used as input for regression models such as LR and RF to predict the cattle's weight.

This feature selection process significantly improves the accuracy of the weight prediction models by removing irrelevant features and focusing only on the most predictive attributes.

## 3.3 Applied CNNs and traditional machine learning algorithms

In this section, we explore three distinct approaches to predict the weight of cattle from images. The first approach leverages a customized CNNs to predict weight directly from processed image data. The second approach integrates EfficientNetB3, a state-of-the-art deep learning architecture, which is fine-tuned to enhance weight prediction performance using transfer learning. The third approach employs traditional machine learning algorithms such as LR and RFR, utilizing features extracted from images through a feature selection process. For all approaches, the dataset was split into 70% for training, 20% for testing, and 10% for validation to ensure reliable model evaluation. These methods were implemented to compare their predictive accuracy and effectiveness in estimating cattle weight. Such automated approaches are essential for advancing precision agriculture by streamlining livestock monitoring and management.

 

**Customized CNNs architectures:** In this research, a customized CNNs was used to predict cattle weight from image data. Each layer of the CNNs architecture plays a crucial role in transforming the input images into meaningful feature representations, which are then used for regression tasks. This research explores four distinct architectures of CNNs models. Fig 6 illustrates four distinct architectures. Specifically, Fig 6(a), 3Conv3Dense, includes three Convolutional layers accompanied by three Dense layers, while Fig 6(b), 3Conv2Dense, comprises three Convolutional layers and two Dense layers. Fig 6(c), 2Conv3Dense, includes two Convolutional layers along with three Dense layers, whereas Fig 6(d), 2Conv2Dense, consists of two Dense layers and two Convolutional layers. All model coefficients are optimised by minimising the MSE between predicted and actual weights. The model's interpretability facilitates comprehension of the impact of each feature on the target variable (cattle's weight). This approach offers a baseline for subsequent comparative analysis with more complex models. Below is a detailed explanation of how each component in the CNNs architecture affects cattle weight prediction.

### 3.3.1 Input layer.

The input to the CNN consists of images with dimensions $(img\_height, img\_width, 3)$, where height and width are predefined image dimensions, and the number 3 represents the RGB color channels. The images are preprocessed (image-to-array conversion, normalization and histogram equalization) to ensure consistency and improve the quality of the data fed into the network. This ensures that the model receives uniform, high-quality input, which is crucial for learning accurate visual patterns related to cattle weight, such as body structure and proportions.

### 3.3.2 Convolutional layers.

The convolutional layers are responsible for automatically extracting important visual features from the input image by applying filters (kernels) [41]. These layers capture both low-level features, such as edges, and high-level features, such as body shapes, which are crucial for predicting cattle weight.

- **First convolutional layer.** The first convolutional layer applies 32 filters of size (3,3) with the ReLU (Rectified Linear Unit) activation function [42]. This layer detects simple patterns, such as edges and textures, that correspond to key structural elements of the cattle's body. The ReLU activation function introduces non-linearity, allowing the model to learn complex relationships between features [43].

   Mathematically, the convolution operation can be represented as:

$$Z^{(1)} = W^{(1)} * X + b^{(1)} \tag{10}$$

   Where:
   - $Z^{(1)}$ is the output of the first convolutional layer,
   - $W^{(1)}$ is the set of learned filters,
   - $X$ is the input image,
   - $b^{(1)}$ is the bias term.

   The ReLU activation function is applied to the output:

$$A^{(1)} = \max(0, Z^{(1)}) \tag{11}$$

   This step enables the network to retain only the positive values from the feature maps, improving the model's ability to handle complex variations in the cattle's physical appearance.

- **Second convolutional layer.** The second convolutional layer applies 64 filters of size (3,3), again with ReLU activation. This layer builds on the simple features detected in the previous layer by identifying more complex patterns, such as curves and shapes that are specific to different parts of the cattle's body. These higher-level features are critical for determining the cattle's weight, as they capture important details about body mass and structure.

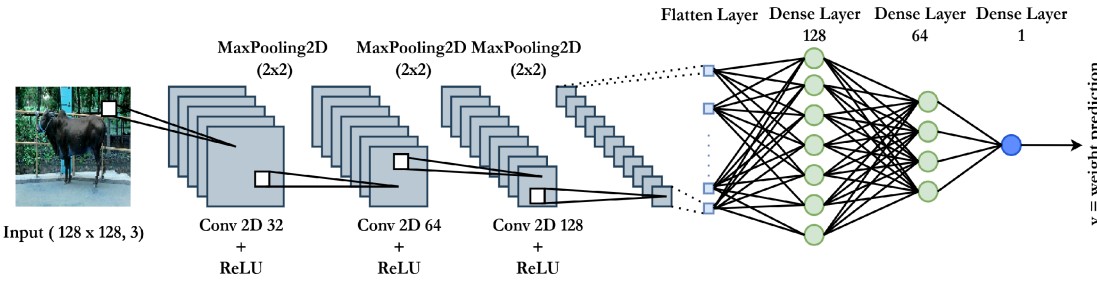

**(a) 3 Convolutional Layers and 3 Dense Layers (3Conv3Dense).**

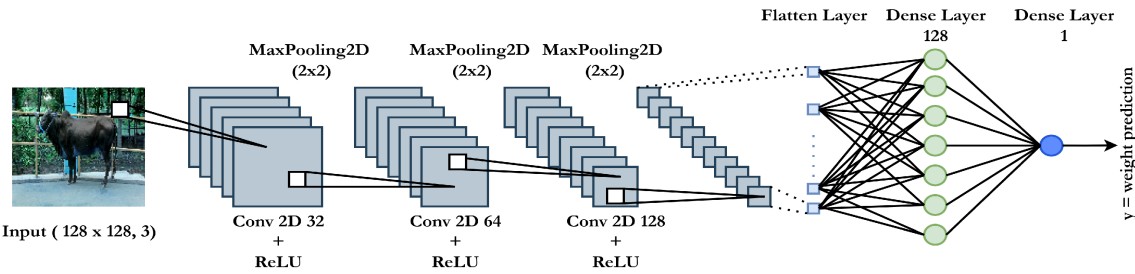

**(b) 3 Convolutional Layers and 2 Dense Layers (3Conv2Dense).**

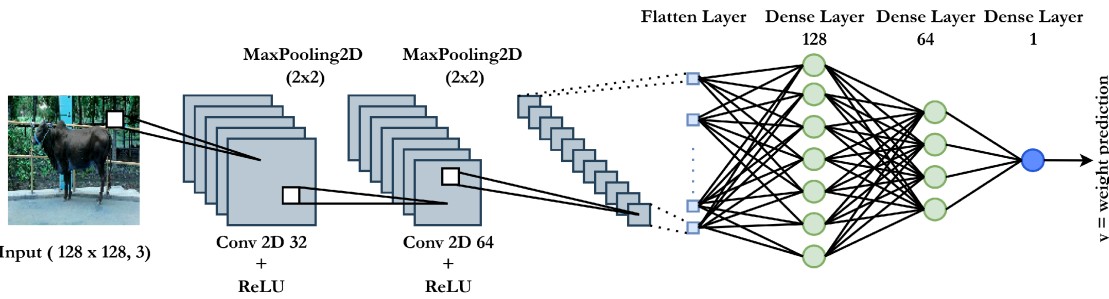

**(c) 2 Convolutional Layers and 3 Dense Layers (2Conv3Dense).**

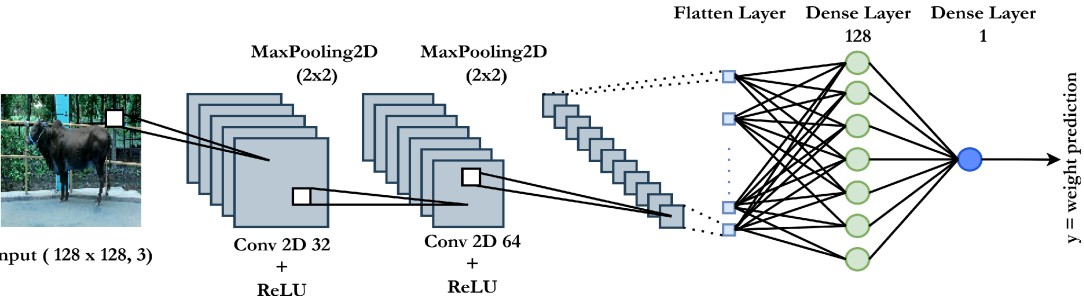

**(d) 2 Convolutional Layers and 2 Dense Layers (2Conv2Dense).**

**Fig 6**. **This figure illustrates the four distinct custom Convolutional Neural Network (CNN) architectures evaluated in this study.** Each diagram showcases a unique configuration, highlighting the variations in the number of convolutional and dense layers designed to assess the impact of architectural depth on the model's performance in predicting cattle weight.

- **Third convolutional layer.** The third convolutional layer applies 128 filters of size (3,3) with ReLU activation. This layer captures even more abstract features and fine-grained details, such as muscle definition, fat distribution, and body composition, which are strongly correlated with the cattle's weight. By learning these intricate features, the model is better able to predict the weight based on visual cues that are not easily discernible to the human eye.

**3.3.3 Max pooling layers.** Each convolutional layer is followed by a Max Pooling operation, which reduces the spatial dimensions of the feature maps, focusing on the most important features and discarding irrelevant information [44]. This step is crucial for:

- **Reducing computational complexity**: By downsampling the feature maps, the model becomes more efficient and faster to train.
- **Preventing overfitting**: Pooling helps to generalize the model by focusing on dominant patterns rather than noise or small variations in the input data [45].

Mathematically, max pooling can be expressed as:

$$P^{(l)} = \max\{A^{(l)}_{i,j,k} \mid (i,j,k) \in \text{pool\_size}\} \tag{12}$$

Where:

- $P^{(l)}$ is the pooled output at layer $l$,
- pool_size is the size of the pooling window.

By reducing the input size, max pooling enables the model to concentrate on the most salient features discarding unnecessary details.

**3.3.4 Flatten layer.** The Flatten Layer converts the 2D feature maps from the convolutional and pooling layers into a 1D vector. This transformation is necessary to transition from the convolutional layers to the fully connected layers, which operate on a linear input [46]. By flattening the features, the model can learn higher-level relationships between the different parts of the cattle's body and how these relate to the cattle's overall weight.

**3.3.5 Fully connected (dense) layers.** The flattened feature map is passed through a series of fully connected layers to learn complex relationships between the features and the cattle's weight.

- **First dense layer.** This layer consists of 128 units and uses the ReLU activation function. It combines the extracted features in a non-linear way, allowing the model to learn more intricate patterns [47]. By combining visual cues such as the shape of the cattle's body and its distribution of mass, the network is able to infer important relationships that contribute to predicting weight.
- **Second dense layer.** The second dense layer consists of 64 units with ReLU activation. This layer refines the feature representations learned in the previous layer, focusing on the most relevant combinations of features that directly impact the cattle's weight. For example, this layer might emphasize the cattle's width and girth as critical factors in determining its overall mass.

**3.3.6 Output layer.** The Output Layer consists of a single unit, which directly predicts the cattle's weight. Since this is a regression task, no activation function is applied to the output layer. The output of the model is a continuous value representing the estimated weight of the cattle.

Mathematically, the output is calculated as:

$$y_{\text{pred}} = W_{\text{out}} \cdot A_{\text{fc2}} + b_{\text{out}} \tag{13}$$

Where:

- $y_{\text{pred}}$ is the predicted cattle weight,
- $W_{\text{out}}$ and $b_{\text{out}}$ are the learned weights and biases for the output layer,
- $A_{\text{fc2}}$ is the activation output from the second dense layer.

Now, Here is a short description on Regression Based ML algorithms applied in our research.

**EfficientNetB3 model architecture:** In this study, we employed EfficientNetB3 to predict cattle weight from cattle's data. While larger variants like EfficientNetB7 are available, we chose EfficientNetB3 because it offers an optimal balance between model complexity, computational efficiency, and predictive accuracy, specially when working with relatively small and diverse datasets like ours. EfficientNetB3 applies a compound scaling method that adjusts depth, width, and resolution in a balanced way, which makes it well suited to extract meaningful features from cattle images without requiring extensive computational resources. Although our custom CNN models ultimately outperformed EfficientNetB3 in weight prediction, EfficientNetB3 served as a strong baseline due to its proven performance on visual tasks and its efficiency in real-world environment. Here in Fig 7, illustrates the implemented EfficientNetB3 model architecture used in our study.

In our implementation, we adopt a transfer learning approach by leveraging a pre-trained EfficientNetB3 model trained on the ImageNet dataset. The top classification layer is removed by setting `include_top=False`, allowing the model to be repurposed for a regression task. This strategy retains the general-purpose convolutional filters learned from large-scale natural images while enabling task-specific learning through a custom regression head. The custom head is composed of:

- **Global average pooling:** Reduces the spatial dimensions of the extracted feature maps into a single feature vector.
- **Dense layers:** Two fully connected layers with 128 and 64 ReLU-activated units for learning complex representations.
- **Output layer:** A single neuron without activation for producing a continuous value representing predicted cattle weight.

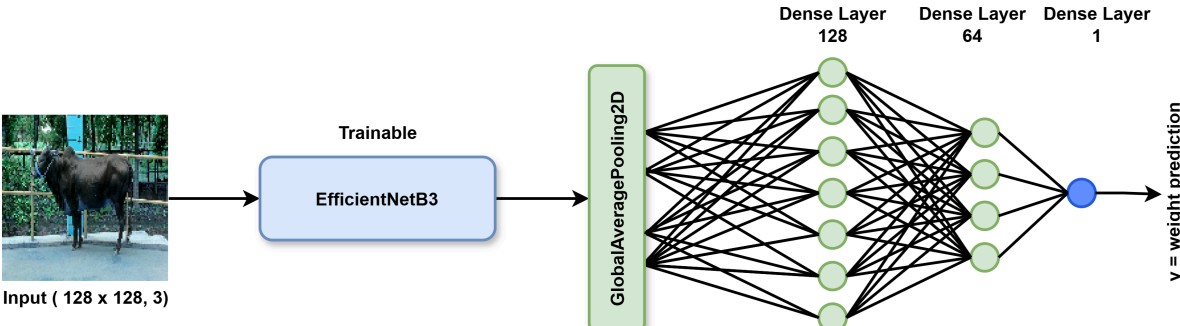

**Fig 7**. **Architecture of the EfficientNetB3-based regression model used for cattle weight prediction.** The pre-trained EfficientNetB3 model is used as a feature extractor, followed by global average pooling and fully connected layers for regression.

The final model outputs a prediction $\hat{y}$ for an input image $I$ using the following formulation:

$$\hat{y} = W_2 \cdot \sigma\left(W_1 \cdot f_\theta(I) + b_1\right) + b_2 \tag{14}$$

here:

- $f_\theta(I)$ is the output feature vector from the EfficientNetB3 base model with parameters $\theta$.
- $W_1, b_1$ are the weights and biases of the first dense layer (128 units).
- $W_2, b_2$ are the weights and biases of the second dense layer (64 units) connected to the final output.
- $\sigma(\cdot)$ denotes the ReLU activation function.

The base model was initially pre-trained and then fine-tuned (`trainable=True`) to adapt to the specific visual patterns in the cattle dataset. This approach ensures that low-level visual features are preserved while allowing high-level features to adjust to domain-specific data. The model was trained using MAE as the loss function to directly optimize for accurate weight prediction.

**Linear regression.** Linear Regression is employed as a fundamental regression technique to model the relationship between the extracted features and the cattle weight. The objective is to establish a linear relationship between the feature and the predicted weight. The general form of the LR model is expressed as:

$$y_{\text{pred}} = \beta_0 + \beta_1 x_1 + \beta_2 x_2 + \cdots + \beta_n x_n \tag{15}$$

Where:

- $y_{\text{pred}}$ represents the predicted cattle weight,
- $\beta_0$ denotes the intercept,
- $\beta_1, \beta_2, \ldots, \beta_n$ are the coefficients corresponding to each feature's contribution,
- $x_1, x_2, \ldots, x_n$ represent the extracted features.

**Random forest regression:** To handle non-linear relationships and interactions between features, we applied RFR. This ensemble learning technique constructs multiple decision trees during training and aggregates their predictions, enhancing the robustness and accuracy of the model [48]. Unlike LR, RFR effectively captures complex, non-linear interactions between features and the target variable.

The final prediction of the RFR model is computed as the average of the predictions from all individual trees:

$$\hat{y}_{\text{pred}} = \frac{1}{T} \sum_{t=1}^{T} f_t(X) \tag{16}$$

Where:

- $\hat{y}_{\text{pred}}$ is the predicted cattle weight,
- $T$ is the number of decision trees,
- $f_t(X)$ is the prediction from the $t$-th decision tree based on the input features $X$.

The RF approach is particularly advantageous when there are intricate, non-linear interactions between the features, which may not be captured by simpler models. By averaging the outputs of several decision trees, the model reduces overfitting and increases the overall prediction accuracy, making it suitable for complex regression tasks in this context.

## 3.4 Hyperparameter tuning and optimization

To ensure optimal and fair evaluation across all models, we applied targeted hyperparameter tuning strategies to both deep learning and traditional machine learning algorithms used in this study. Our experiments involved a total of seven models: four custom CNN architectures, EfficientNetB3, Linear Regression, and Random Forest Regression.

**Deep learning models.**

The following hyperparameters were applied uniformly across the five deep learning-based models (four custom CNNs and EfficientNetB3), selected based on empirical performance observed on the validation set using MAE and MAPE as primary evaluation metrics.

- Learning rate: A learning rate of 0.0001 was chosen to enable gradual convergence and reduce the risk of overshooting during training.
- Batch size: A batch size of 40 was selected to balance computational efficiency and convergence stability.
- Dropout rate: A dropout rate of 0.03 was applied to prevent overfitting by randomly deactivating neurons during training. Dropout can be mathematically represented as:

$$h_{\text{drop}}^{(l)} = m^{(l)} \odot h^{(l)}, \quad m^{(l)} \sim \text{Bernoulli}(1-p) \tag{17}$$

where $h^{(l)}$ is the output of layer $l$, $m^{(l)}$ is the dropout mask, $p$ is the dropout probability, and $\odot$ denotes element-wise multiplication.
- L2 Regularization: An L2 regularization coefficient of $\lambda = 0.001$ was used to penalize large weight values and enhance generalization. This adds a penalty term to the loss function, defined as:

$$\text{Loss}_{\text{L2}} = \text{Loss}_{\text{original}} + \lambda \sum_{j=1}^{m} w_j^2 \tag{18}$$

where $w_j$ represents the model weights and $m$ is the total number of weights.

These hyperparameter choices contributed significantly to the strong performance of our CNN models, particularly the 3Conv3Dense architecture, which achieved the best MAE and MAPE scores across all experiments.

**Traditional machine learning models**

For the traditional machine learning models, we optimized the RFR by setting the number of estimators to 100 (n_estimators=100) and the random_state to 42 to ensure reproducibility. This configuration was selected after evaluating a range of parameter values, in order to balance predictive performance with computational efficiency. The RFR was applied to the feature-extracted dataset obtained through YOLOv5 and refined using RFE. In contrast, the Linear Regression model was used with default parameters, as it does not require tuning beyond feature preprocessing. RFE was also applied prior to linear regression to reduce dimensionality and retain the most informative features for weight prediction.

To ensure a robust evaluation of both models, we employed 10-fold cross-validation, which involves partitioning the dataset into ten equal subsets. Each model was trained on nine subsets and validated on the remaining one, iteratively, to obtain an average performance metric. This approach mitigates the risk of biased evaluation due to data partitioning

Together, these optimization steps ensured that each of the seven models was trained under appropriate and well-tuned conditions, supporting a fair and rigorous performance comparison.

# 4 Result analysis

## 4.1 Evaluation metrics

Here is the result analysis section. The results have been evaluated using various metrics such as MAE, MSE, RMSE, and $R^2$ values. Additionally, we have visualized the results with different charts and graphs for better clarity. Detailed explanations are provided below.

**4.1.1 Mean Absolute Error (MAE).** Mean Absolute Error (MAE) [13] is the average of the absolute differences between the predicted and actual values. It gives an indication of the average magnitude of errors in a set of predictions, without considering their direction. The formula for MAE is:

$$\text{MAE} = \frac{1}{n} \sum_{i=1}^{n} |y_i - \hat{y}_i| \tag{19}$$

Where:

- $n$ is the number of samples,
- $y_i$ is the actual value (true weight),
- $\hat{y}_i$ is the predicted value (predicted weight).

MAE provides a straightforward interpretation—on average, how far off the predictions are from the actual values. In this research, for example, the CNN model achieved a MAE of 18.01 kg, meaning that, on average, the predicted cattle weights are 18.01 kg away from the true weights. This highlights the capability of the deep learning model in minimizing prediction errors.

**4.1.2 Mean Absolute Percentage Error (MAPE).** MAPE, measures the average magnitude of the errors between predicted and actual values, expressed as a percentage of the actual values. It is particularly useful when the interpretability of error in percentage terms is desired. The formula for MAPE is:

$$\text{MAPE} = \frac{100}{n} \sum_{i=1}^{n} \left| \frac{y_i - \hat{y}_i}{y_i} \right| \tag{20}$$

MAPE provides an intuitive understanding of prediction accuracy by indicating how far predictions deviate from actual values on average, relative to the true values. Unlike MSE, it does not penalize larger errors more heavily, making it less sensitive to outliers. Lower MAPE values indicate better model performance. In this study, MAPE serves as a key metric for evaluating the accuracy of cattle weight predictions, offering a clear, percentage-based error measure across different weight ranges.

**4.1.3 Mean Squared Error (MSE).** Mean Squared Error (MSE) [49] is the average of the squared differences between the actual and predicted values. Squaring the errors penalizes larger errors more than smaller ones, making MSE useful when large errors are particularly undesirable. The formula for MSE is:

$$\text{MSE} = \frac{1}{n} \sum_{i=1}^{n} (y_i - \hat{y}_i)^2 \tag{21}$$

The squaring of the error terms makes MSE sensitive to outliers, meaning large prediction errors will have a higher impact on the overall MSE value. Lower MSE values indicate better performance. In this study, the model's MSE helps to understand how well the model performs on average, accounting for large discrepancies between predictions and actual values.

 

**4.1.4 Root Mean Squared Error (RMSE).** Root Mean Squared Error (RMSE) [13] is the square root of the MSE. It is used to restore the units of the error to the same units as the target variable (in this case, kilograms). RMSE is often more interpretable than MSE because it is in the same units as the predicted quantity. The formula for RMSE is:

$$\text{RMSE} = \sqrt{\frac{1}{n}\sum_{i=1}^{n}(y_i - \hat{y}_i)^2} \tag{22}$$

RMSE is particularly useful when comparing model performances since it penalizes larger errors more severely than MAE, but still provides a result in the same unit as the predicted values (kg). A lower RMSE value would indicate a model that makes fewer large errors, making it preferable in cases where large deviations in predictions are unacceptable.

**4.1.5 $R^2$ Value:** The $R^2$ value, or coefficient of determination, measures how well the predicted values align with the actual values [50]. It evaluates the proportion of variance in the target variable in our case the cattle's weight that is explained by the model using the input features which is cattle's image. A higher $R^2$ value signifies a stronger fit, meaning the model reliably captures the relationship between the images and the corresponding weights. The formula of $R^2$ is defined as:

$$R^2 = 1 - \frac{\sum_{i=1}^{n}(y_i - \hat{y}_i)^2}{\sum_{i=1}^{n}(y_i - \bar{y})^2} \tag{23}$$

## 4.2 Result

Table 1, presents a comprehensive comparison of the performance of various algorithms in predicting cattle weight from image data. The evaluation metrics include MAE, MAPE, MSE, RMSE, $R^2$ and training time, enabling both predictive accuracy and computational efficiency to be assessed.

Among all tested models, the Custom CNN architecture with the 3Conv3Dense configuration achieved the best overall performance, recording the lowest MAE (18.02 kg) and MAPE (6.22%), along with the smallest MSE (394.24) and RMSE (19.85). Also, it attained the highest coefficient of ($R^2$ = 94.32%), indicating excellent agreement between predicted and actual weights. Despite its high predictive accuracy, the training time (11,430 ms) remained competitive, only marginally higher than the other CNN configurations. This suggests that the additional convolutional and dense layers contribute to accuracy improvements without imposing excessive computational overhead.

The second-best performer within the CNN group was the 2Conv3Dense configuration, with an MAE of 21.48 kg, MAPE of 7.48%, and $R^2$ of 91.43%. Although it exhibited slightly higher error values than the 3Conv3Dense model, it still surpassed most other models in terms of predictive accuracy. The 3Conv2Dense configuration also showed strong results, achieving a MAPE of 7.61% and $R^2$ of 91.54%, suggesting that the depth of convolutional layers plays a critical role in feature extraction quality. The 2Conv2Dense architecture, while still competitive, recorded the highest errors

**Table 1**. MAE, MAPE, MSE, RMSE, $R^2$ and Training Time (ms) for different algorithms utilised in this research.

| Algorithm | Architecture | MAE | MAPE (%) | MSE | RMSE | $R^2$ | Training Time (ms) |
|---|---|---|---|---|---|---|---|
| **Custom CNN** | **3Conv3Dense** | 18.02 | 6.22 | 394.24 | 19.85 | 94.32 | 11,430 |
| | **3Conv2Dense** | 22.32 | 7.61 | 545.76 | 23.36 | 91.54 | 11,420 |
| | **2Conv3Dense** | 21.48 | 7.48 | 476.33 | 21.83 | 91.43 | 10,340 |
| | **2Conv2Dense** | 24.03 | 8.28 | 621.45 | 24.92 | 88.56 | 10,200 |
| **EfficientNetB3** | — | 21.05 | 7.25 | 446.71 | 21.13 | 91.32 | 90,500 |
| **Random Forest** | — | 23.67 | 8.16 | 833.47 | 28.87 | 78.56 | 87,000 |
| **Linear Regression** | — | 25.99 | 8.96 | 855.36 | 29.24 | 72.13 | 82,300 |

among the CNN group (MAE = 24.03 kg, MAPE = 8.28%) and a lower $R^2$ of 88.56%, reflecting reduced capacity to capture complex visual patterns when both convolutional and dense layers are reduced.

The EfficientNetB3 model, used as a strong pre-trained benchmark, achieved an MAE of 21.05 kg and a MAPE of 7.25%. It performed better than both Random Forest and Linear Regression but did not surpass the top-performing custom CNN architectures. Although EfficientNetB3 benefits from advanced compound scaling and ImageNet pretraining, its general-purpose design, mainly optimized for classification tasks, likely limited its performance in this regression-focused problem without further domain-specific fine-tuning. Moreover, it had the longest training time (90,500 ms) among all deep learning models, indicating a trade-off between its complexity and computational efficiency.

Traditional ML methods, such as Random Forest Regression and Linear Regression, performed considerably worse than DL approaches. Random Forest achieved an MAE of 23.67 kg and MAPE of 8.16%, with an $R^2$ of 78.56%, while Linear Regression fared the worst overall, with an MAE of 25.99 kg, MAPE of 8.96%, and the lowest $R^2$ of 72.13%. Both models also required significantly longer training times (87,000 ms for Random Forest and 82,300 ms for Linear Regression) compared to CNN architectures, highlighting their inefficiency in processing high-dimensional image features.

In summary, the results clearly demonstrate that CNN architectures, particularly the 3Conv3Dense CNN, provide the most effective balance between predictive accuracy and computational cost. While EfficientNetB3 offers competitive performance as a pre-trained model, its lack of domain-specific adaptation limits its ability to outperform a tailored CNN architecture. Classical regressors, though computationally simpler in theory, proved less effective and less efficient for this high-dimensional visual prediction task.

### 4.3 Actual vs predicted and residual graphs

Figs 8 and 9 illustrate the Actual vs. Predicted and Residual Graphs, offering a comprehensive assessment of the models' performance through a visual comparison of predicted cattle weights with actual values and an analysis of the distribution of prediction errors (residuals). These graphs are crucial for assessing the usefulness of each model in relating to new data, especially across a diverse range of cattle weights.

The Actual vs. Predicted graphs evaluate each model's performance based on the alignment of data points along the diagonal line, indicative of perfect prediction accuracy. The 3Conv3Dense CNN design Fig 8(a), exhibits the highest prediction accuracy, with data points closely aligned along the diagonal. This consistency is particularly visible at higher weights, where prediction mistakes are greater in other models. The 3Conv3Dense model's deeper design, with three convolutional and three dense layers, certainly improves its ability to collect complex, non-linear patterns in image data, making it more proficient at handling diverse visual qualities such as body shape, size, and texture.

Conversely, the 3Conv2Dense architecture Fig 8(b), exhibits a little rise in inaccuracy as weight values grow, suggesting that the reduction in dense layers compromises the model's ability to effectively integrate learnt features. The 2Conv3Dense Fig 8(c) and 2Conv2Dense Fig 8(d), models exhibit increasing divergence from the diagonal, particularly for heavier cattle, suggesting that their shallower architectures constrain their ability to capture complex relationships within the data. This underscores the significance of deeper networks in image-based regression tasks, where the intricacy of the input data necessitates a more sophisticated feature extraction approach.

The EfficientNetB3 model, Fig 8(e) demonstrates competitive performance, with Actual vs. Predicted values aligning closely along the diagonal. While not outperforming the 3Conv3Dense architecture, EfficientNetB3 displays notable consistency across a wide range of cattle weights. Its pre-trained and compound-scaled architecture allows it to efficiently balance depth, width, and resolution, thereby capturing intricate visual patterns from cattle images.

Conventional machine learning models, like RFR Fig 8(f) and LR Fig 8(g), exhibit markedly inferior performance. The RFR model demonstrates satisfactory performance at lower weights but displays increased variance at higher weights, suggesting difficulty in generalising to more intricate data points. This is probably attributable to the model's dependence on decision trees, which may inadequately represent the complex, non-linear relationships inherent in image data.

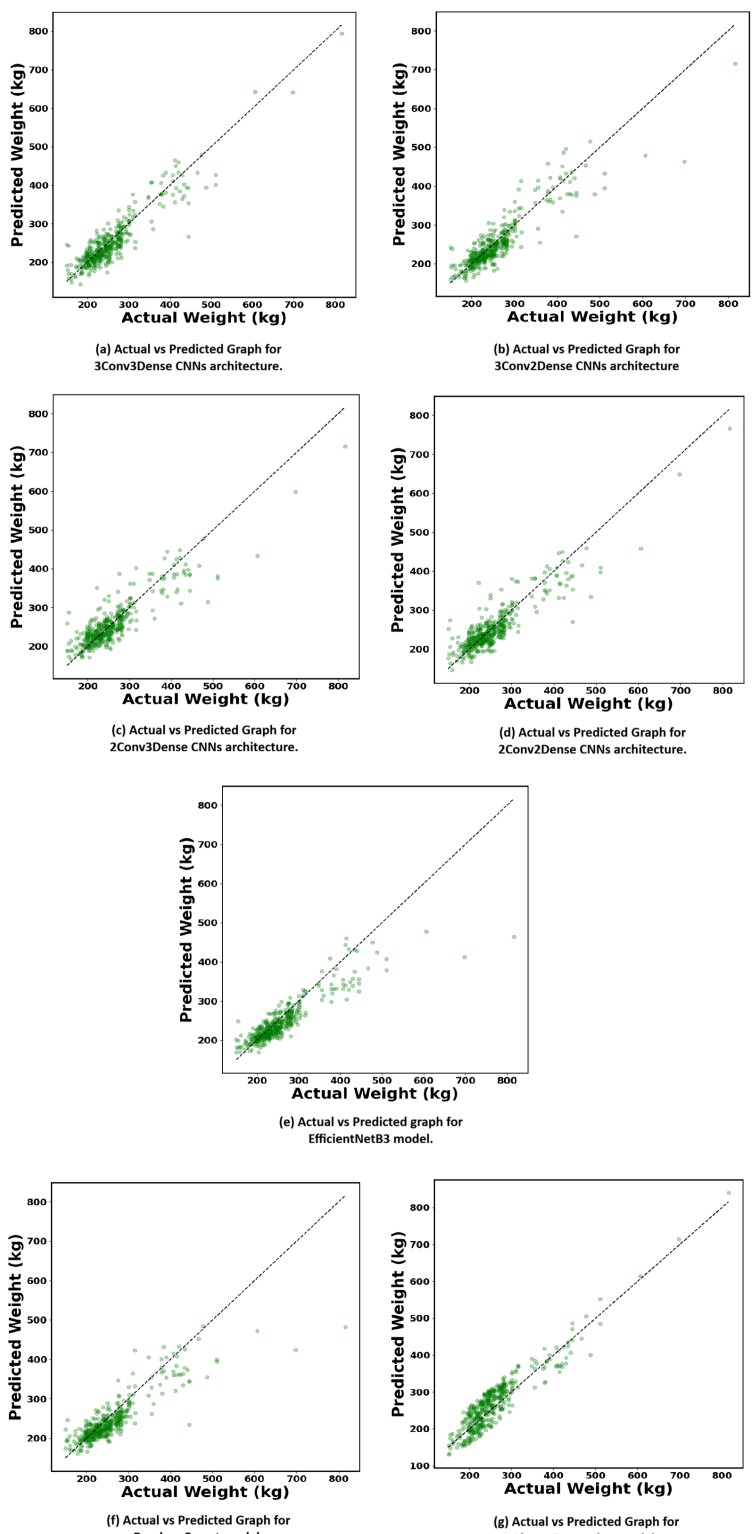

(a) Actual vs Predicted Graph for
3Conv3Dense CNNs architecture.

(b) Actual vs Predicted Graph for
3Conv2Dense CNNs architecture

(c) Actual vs Predicted Graph for
2Conv3Dense CNNs architecture.

(d) Actual vs Predicted Graph for
2Conv2Dense CNNs architecture.

(e) Actual vs Predicted graph for
EfficientNetB3 model.

(f) Actual vs Predicted Graph for
Random Forest model.

(g) Actual vs Predicted Graph for
Linear Regression model.

**Fig 8**. **Comparison of Actual vs. Predicted values across different models used in this study.** The graphs illustrate the performance of each model in predicting cattle weight and highlighting how closely the predicted values align with the actual weights.

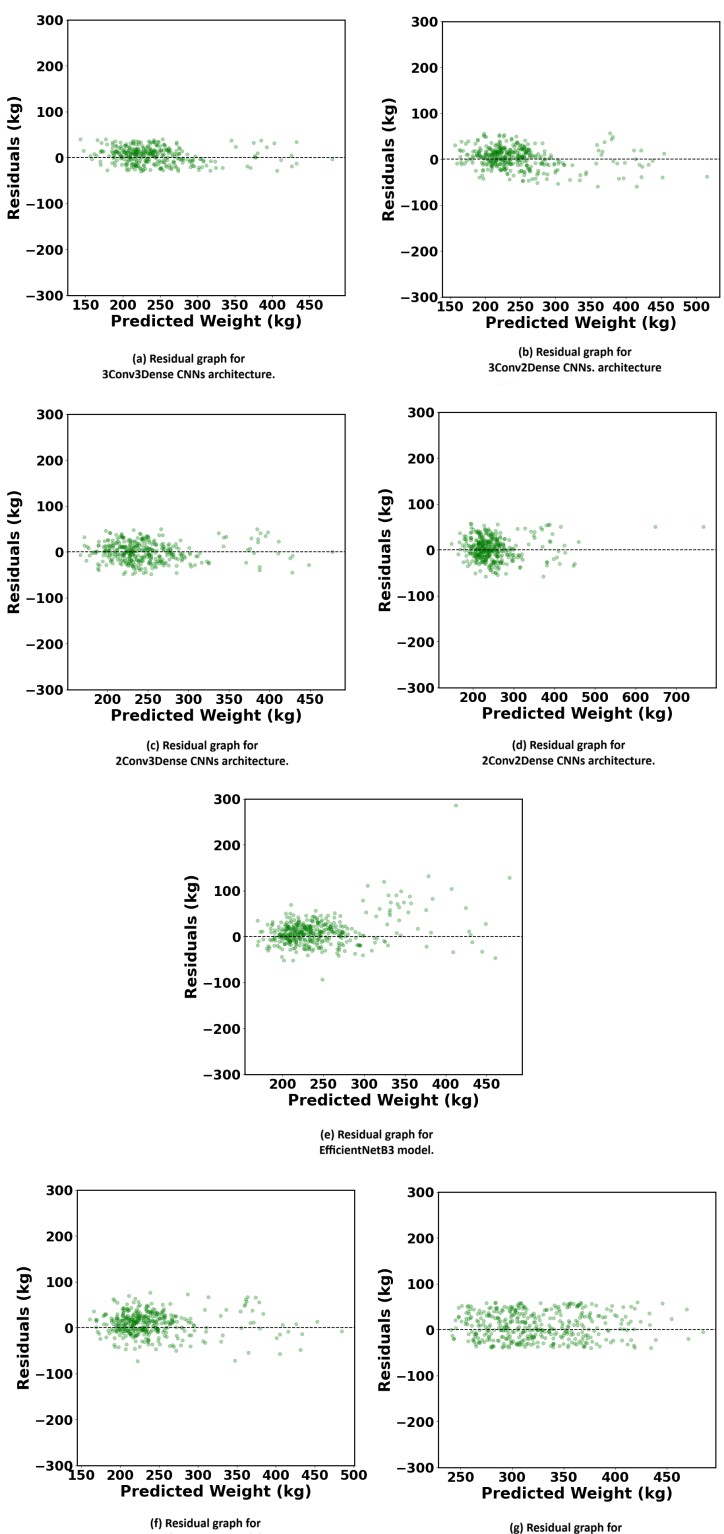

**Fig 9. Visualization of Residuals across different models applied in our research.** The residual plots help assess model performance by highlighting deviations between actual and predicted weights.

The LR model, as anticipated, has the poorest performance, characterised by a significant dispersion of points distant from the optimal line, particularly for heavier cattle. This indicates that the linear model is inappropriate for this task, since it fails to accurately capture the non-linearities contained in image data, resulting in consistent under and overestimation.

The Residual Graphs further validate these findings. In an optimal model, residuals denoting the disparity between actual and predicted value ought to be randomly distributed around zero, signifying the absence of systematic bias in the model. The 3Conv3Dense CNN architecture Fig 9(a), generates residuals that are closely grouped around zero, indicating its constant accuracy in predicting cattle weights with minimum error. The equal distribution of residuals across all weight ranges underscores the model's excellent generalisation, rendering it extremely appropriate for practical applications in livestock management where precision is essential.

The 3Conv2Dense Fig 9(b) and 2Conv3Dense Fig 9(c) models exhibit heightened residual variance, especially at elevated weights, suggesting that their diminished complexity constrains their predictive precision. The 2Conv2Dense model Fig 9(d), demonstrates considerable over- and under-predictions, indicating that its superficial architecture is inadequate for capturing the intricate visual patterns necessary for precise weight prediction.

On the other hand, EfficientNetB3 model maintains a relatively compact residual distribution, as depicted in Fig 9(e). Most residuals are concentrated near the zero-error line, with only a few moderate deviations. This suggests that prediction errors are not only smaller but also less systematically biased across the weight spectrum. Compared to shallower networks like 2Conv2Dense or 2Conv3Dense, the EfficientNetB3 model shows improved stability and lower variance in predictions, especially in the mid-to-high weight ranges where prediction complexity typically increases.

The conventional models, RF Fig 9(f) and LR Fig 9(g), underscore the inadequacies of simpler algorithms for this purpose. The residuals of the RF model exhibit greater dispersion, particularly at elevated weights, indicating larger and more unpredictable prediction mistakes. This indicates that although RF can manage simpler linkages in the data, it encounters difficulties with the complex, non-linear dependencies inherent in image-based weight calculation. The LR model exhibits the most pronounced aberrations, with residuals displaying a clear trend of underestimating bigger weights and overestimating lighter weights. This recurrent bias highlights the inadequacy of linear models for intricate tasks such as predicting cattle weight from images, when the connections among variables are fundamentally non-linear.

## 4.4 Validation curves

The validation process for the CNN architectures was crucial in assessing the model's broader performance [51]. Throughout the training process, we assessed the loss function for both the training and validation datasets to determine the model's learning efficacy and its capacity for generalisation beyond the training data. Fig 10, presents the curves of Validation Loss compared to Training Loss for the four distinct CNN architectures examined in this study.

The validation curve of 3Conv3Dense architecture shows in Fig 10(a), demonstrates a consistent decrease in training loss over time, suggesting effective learning of features from the training data. The validation loss exhibits a comparable trend, steadily decreasing with minimal variation. The close alignment of training and validation loss indicates that the model effectively avoids overfitting, showcasing a robust ability to generalise to new data. The stability and convergence of the validation loss curve demonstrate the robustness of this architecture, contributing to its superior performance across various evaluation metrics, including the lowest MAE and RMSE values.

The validation curve of 3Conv2Dense architecture shows in Fig 10(b), demonstrates a larger disparity between training and validation loss, especially in the later phases of training. The convergence of validation loss, coupled with a wider gap, suggests mild overfitting, as the model identifies specific patterns in the training data that do not generalise effectively to the validation set. The overall trend remains positive, and the model exhibits reasonable generalisation capabilities, as indicated by its competitive, though slightly inferior, performance relative to the 3Conv3Dense architecture.

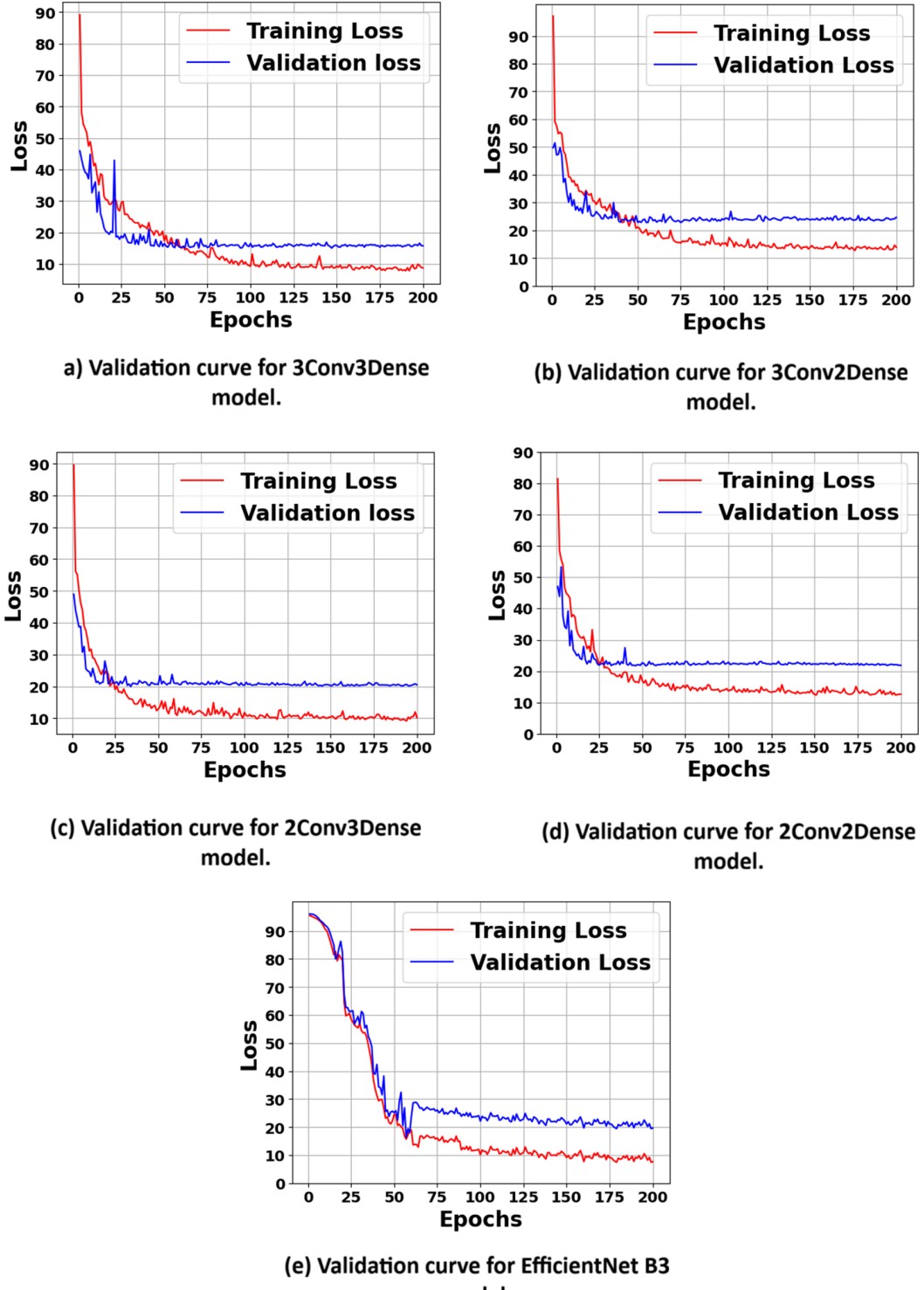

a) Validation curve for 3Conv3Dense model.

(b) Validation curve for 3Conv2Dense model.

(c) Validation curve for 2Conv3Dense model.

(d) Validation curve for 2Conv2Dense model.

(e) Validation curve for EfficientNet B3 model.

**Fig 10. Validation Loss vs. Training Loss Curves for the CNN architectures and EfficientNet B3 model evaluated in this study.** These plots illustrate the learning behavior of each model during training, helping to assess convergence, overfitting, and generalization capability.

The loss curves for both training and validation in the 2Conv3Dense architecture shows in Fig 10(c) exhibit shapes analogous to those observed in the 3Conv3Dense model. The validation loss reaches a plateau earlier, indicating a limitation in the model's capacity for further improvement. This earlier plateau suggests that this architecture is relatively less effective in learning complex features from the image data, as evidenced by its slightly higher MAE and RMSE values.

The validation curves of 2Conv2Dense architecture shows in Fig 10(d), exhibits the most pronounced disparity between training and validation loss. The validation loss stabilises significantly earlier, whereas the training loss persists in its decline, suggesting a notable issue of overfitting. The model's limited generalisation to the validation data leads to suboptimal performance, as evidenced by the highest MAE and RMSE values among the evaluated architectures. The initial plateau and broad gap indicate that this architecture is overly simplistic for accurately representing the complex, non-linear relationships present in the cattle image dataset.

The validation curve of the EfficientNetB3 architecture, as illustrated in Fig 10(e), reveals a rapid convergence of both training and validation loss in the early epochs. The close alignment between the two curves indicates minimal overfitting and excellent generalisation capabilities. This stability across the training process suggests that EfficientNet B3 efficiently extracts relevant features from the image data with fewer fluctuations in loss, even as training progresses. As a result, it delivers consistent performance, reflecting its strong architectural efficiency and transfer learning benefits in handling complex visual features within the cattle image dataset.

## 4.5 Prediction accuracy distribution

Figs 11 and 12, provide a comparative analysis of the performance of different models such as, Custom CNNs, RFR, and LR, in predicting cattle weight. Among the evaluated CNNs architectures, the 3Conv3Dense model demonstrates the most significant and focused distribution of predictions within the 0-10% error range, as illustrated in Fig 11(a). This suggests that most predictions exhibit high accuracy with negligible error margins. The steep distribution curve indicates that this model successfully captures the complex, non-linear relationships inherent in the image data, resulting in enhanced predictive performance.

The improved performance is supported by the Fig 12, which demonstrates that the predicted weights from the 3Conv3Dense CNN closely match the actual values. This close correspondence illustrates the model's robustness and practical applicability in livestock management, especially for precise and automated weight estimation. The architecture comprises three convolutional layers succeeded by three dense layers, facilitating the extraction of sophisticated, structured features from cattle images. The convolutional layers identify critical spatial patterns, including body shape, muscle distribution, and texture, whereas the dense layers enable non-linear integration of these features, enhancing the model's weight predictions.

Alternative CNN architectures, including 3Conv2Dense and 2Conv3Dense, exhibit satisfactory performance; however, they do not achieve the precision levels of the 3Conv3Dense model. The 3Conv2Dense model, while employing an equivalent number of convolutional layers, is constrained by a reduced number of dense layers, leading to less effective feature integration. As a result, the predictions exhibit increased dispersion, with a higher proportion situated within the 10-20% error range, as shown in Fig 11(b). The 2Conv3Dense model, characterized by a reduced number of convolutional layers, exhibits marginally lower accuracy as a result of its diminished ability to extract complex features, resulting in a wider error distribution.

The Random Forest Regression model in Fig 11(f) demonstrates a wider error distribution, with predictions covering a broader range, especially in the 10-20% and 20-30% error intervals. This suggests that although the model demonstrates satisfactory performance, it fails to comprehensively account for the complexities within the data, resulting in increased variability in predictions.

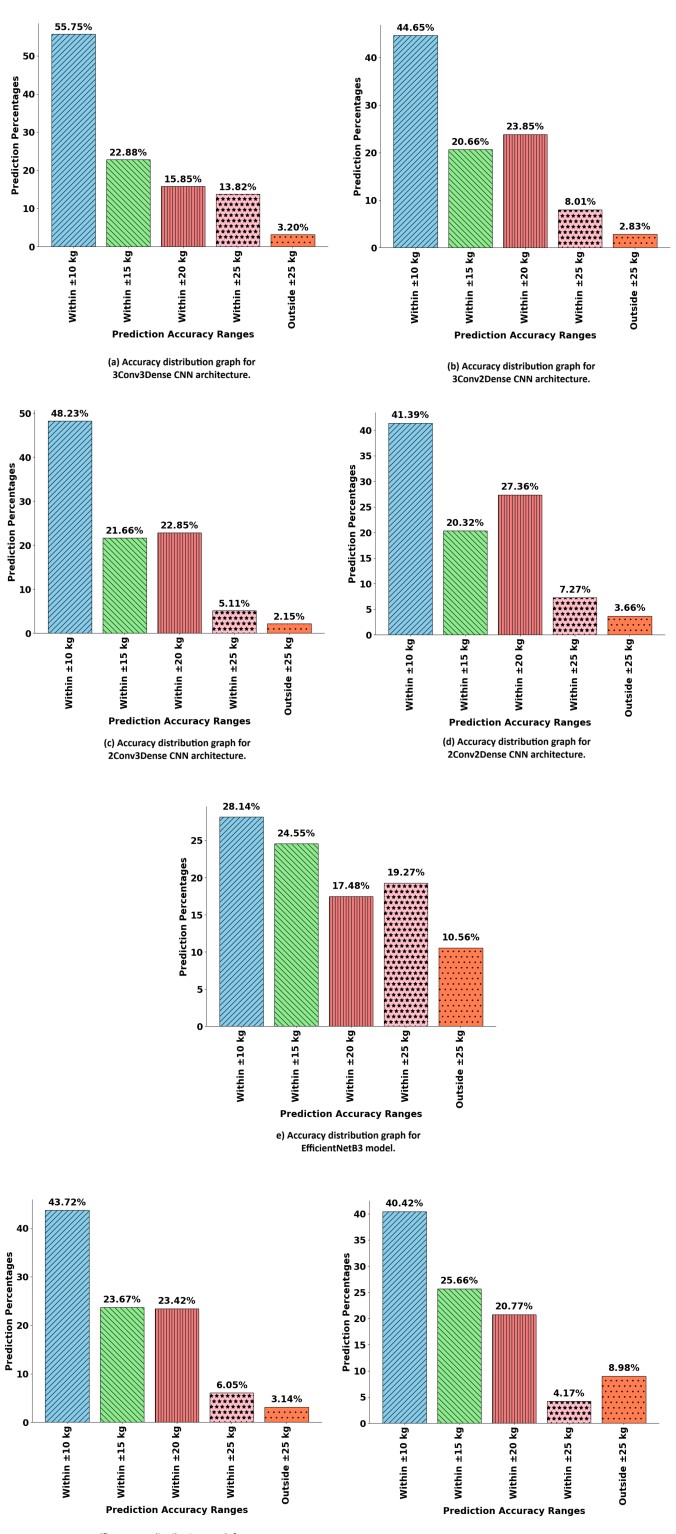

(a) Accuracy distribution graph for 3Conv3Dense CNN architecture.

(b) Accuracy distribution graph for 3Conv2Dense CNN architecture.

(c) Accuracy distribution graph for 2Conv3Dense CNN architecture.

(d) Accuracy distribution graph for 2Conv2Dense CNN architecture.

e) Accuracy distribution graph for EfficientNetB3 model.

(f) Accuracy distribution graph for Random Forest model.

(g) Accuracy distribution graph for Linear Regression model.

**Fig 11**. **Prediction percentage distribution using different algorithms applied in our research.**

Actual: 230.00 kg
Predicted: 233.67 kg
Difference: {3.67} kg

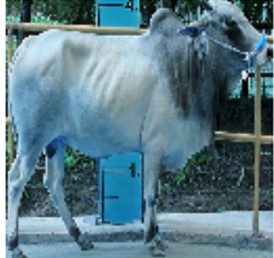

Actual: 275.00 kg
Predicted: 279.68 kg
Difference: {4.68} kg

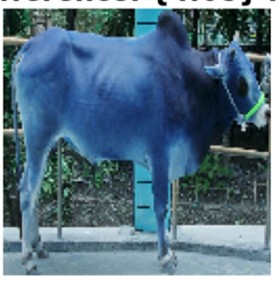

Actual: 289.00 kg
Predicted: 287.29 kg
Difference: {1.71} kg

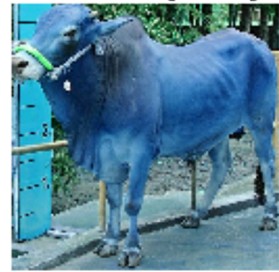

Actual: 300.00 kg
Predicted: 296.49 kg
Difference: {3.51} kg

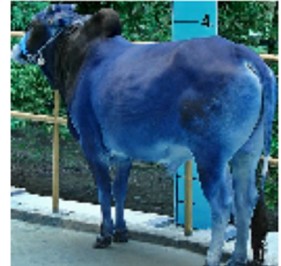

Actual: 170.00 kg
Predicted: 168.19 kg
Difference: {1.81} kg

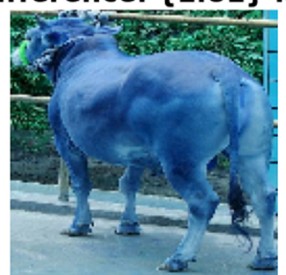

Actual: 300.00 kg
Predicted: 300.49 kg
Difference: {0.49} kg

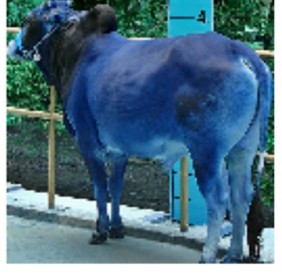

**Fig 12**. **Here are the samples prediction of cattle weight using our proposed best 3Conv3Dense model.**

The Linear Regression model exhibits the lowest performance, as a considerable proportion of predictions exceed the 30% error margin. The extensive distribution highlights the inadequacies of a linear model in addressing the non-linear, high-dimensional nature of image data, leading to significant prediction errors.

The 3Conv3Dense CNN architecture exhibits enhanced prediction accuracy and consistency relative to simpler CNN models and conventional regression methods. The capacity to reduce errors across diverse test cases, particularly when compared to Random Forest and Linear Regression, establishes it as the most dependable method for automated livestock weight estimation. This model is highly appropriate for implementation in precision agriculture, where precise and scalable solutions are critical for real-time livestock management.

On the other hand, EfficientNetB3 model, as illustrated in Fig 11(e), exhibits a relatively balanced distribution of prediction accuracy across multiple error ranges. Although only 28.14% of the predictions fall within the 0–10% error range, which is lower than the top-performing 3Conv3Dense model, the results remain fairly consistent across the 10–30% error range. Specifically, 24.55% fall between 10–15%, 17.46% between 15–20%, and 19.27% between 20–30%. However, about 10.56% of the predictions exceed a 30% error margin, which is indicating moderate variability in the model's performance. This distribution suggests that EfficientNetB3 is capable of generalizing well across the dataset but may not match the precision of more tailored CNN architectures in capturing domain-specific features crucial for livestock weight estimation. Nonetheless, the model's robustness and scalability make it a viable option for deployment in practical scenarios where training efficiency and model transferability are prioritized.

The improved performance of the 3Conv3Dense architecture arises from its capacity to effectively capture both low-level and high-level features in cattle images. The three convolutional layers extract spatial hierarchies, encompassing body shape, muscle distribution, and texture, while the three dense layers non-linearly integrate these features to enhance weight predictions. This architectural combination enables the model to learn intricate correlations between visual data and cattle weight, surpassing the effectiveness of traditional regression models and simpler CNN architectures. The correlation between predicted and actual values demonstrates the model's capacity to generalize across varied samples, reinforcing its designation as the most appropriate model for real-world applications necessitating high precision in livestock weight estimation.

In Fig 12, we can see that the predictions generated by the 3Conv3Dense model for several randomly selected test cattle images closely align with the actual weights. This strong correspondence highlights the model's effectiveness in accurately estimating cattle weight from image data, underscoring its potential applicability in real-world livestock monitoring scenarios.

Table 2, presents a comparison of various approaches for estimating cattle weight, employing MAE as a metric for accuracy. The custom CNN model, namely the "3Conv3Dense" variant shown in our study, attained optimal performance with the minimal MAE of 18.02. The low error rate underscores the model's potential as a more precise and scalable approach for calculating cattle weight non-invasively, rendering it advantageous for agricultural management.

## 4.6 Interpreting model decision and error case analysis with lime

In this section, we employed LIME to examine the decision-making process of our proposed CNNs models and conduct error case analysis. LIME visualizations offered information about the specific input attributes that influenced the model's weight projections.

Our visual analysis revealed that the model consistently focused on anatomically relevant regions, including the rib cage, abdomen, and hindquarters, areas known to reflect muscle mass and fat distribution, which are critical indicators of weight. As shown in Fig 13, these regions are prominently highlighted in red, indicating a strong positive contribution to the model's weight estimation. The high consistency of these highlighted areas across different samples suggests that the CNN learned meaningful visual representations aligned with expert intuition.

To evaluate the robustness of our model, we analysed error cases shown in Fig 14. In Fig 14(a), exhibited a large prediction deviation, where the actual weight was 252 kg, but the 2Conv2Dense model predicted 234.33 kg, demonstrating significant underestimation. However, the 3Conv3Dense model, with its deeper architecture, successfully reduced the error in a similar scenario in Fig 14(b), where the actual weight was 252 kg, and the prediction was 253.66 kg, showcasing improved accuracy. These findings validate the efficacy of our proposed model in minimizing errors and highlight its superior ability to capture intricate, non-linear patterns in image data. This interpretative approach assures that the framework is not a "black-box" solution. It also supports trust and usage among agricultural practitioners by providing both accuracy and explainability in cattle weight estimation.

**Table 2**. **Performance comparison of our proposed model with existing approaches for cattle weight prediction using machine learning and deep learning techniques.**

| Article | Best Method | Number of Cattle | Best MAE |
|---|---|---|---|
| [27] | CNN | 20 | 23.19 |
| [52] | Linear Regression | 34 | 38.46 |
| [22] | MRGBDM | 154 | 35.5 |
| [31] | ExtraTreesRegressor | 1500 | 25.19 |
| [25] | RandomForestRegressor | 270 | 25.2 |
| **Our Best Proposed Model** | **Conv3Dense3** | **513** | **18.02** |

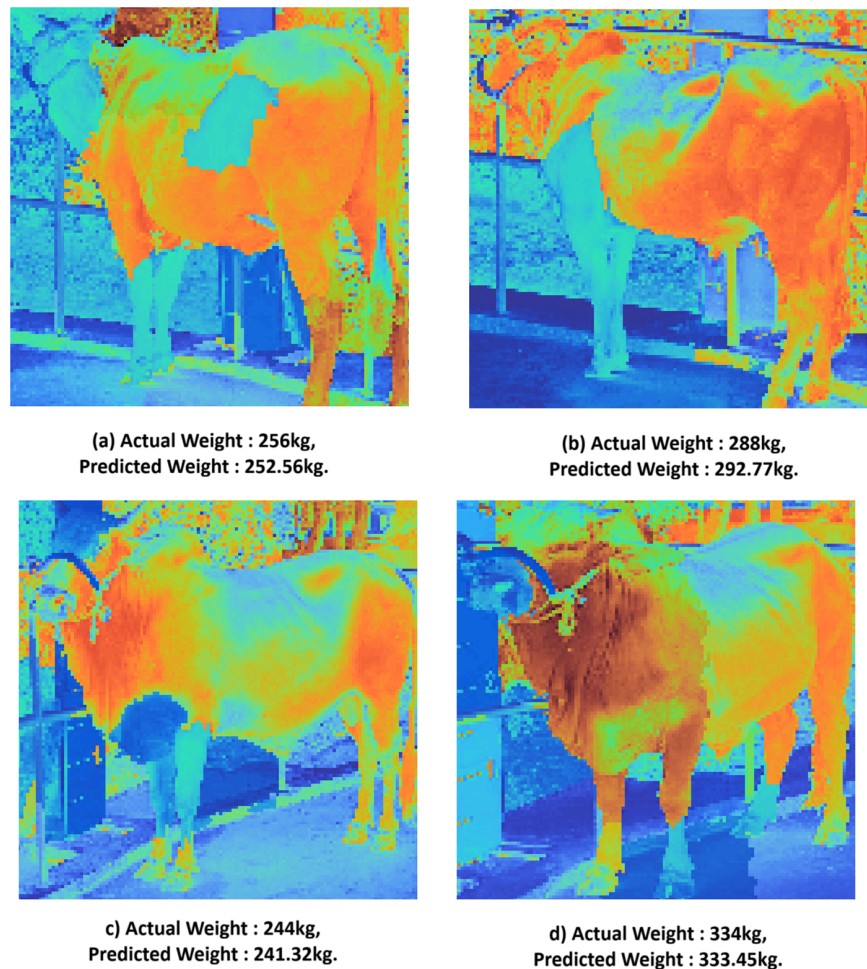

(a) Actual Weight : 256kg,
Predicted Weight : 252.56kg.

(b) Actual Weight : 288kg,
Predicted Weight : 292.77kg.

c) Actual Weight : 244kg,
Predicted Weight : 241.32kg.

d) Actual Weight : 334kg,
Predicted Weight : 333.45kg.

**Fig 13**. **LIME visualizations for the 3Conv3Dense model, providing insight into its prediction process.** The highlighted regions indicate the key visual features that contributed most to the weight estimation. The visualizations show the model consistently focusing on anatomically relevant areas, such as the rib cage and abdomen.

## 5 Limitations

Here are the limitations of our research. First, the dataset used includes only cattle breeds from Bangladesh, so applying our model to breeds from other regions is still unexplored. Additionally, no separate analysis was conducted to evaluate prediction accuracy across different breeds, which could influence results due to breed-specific morphological variations.

Moreover, most cattle in the dataset weigh between 200 and 400 kg, and our model performed extremely well for this range. However, for cattle weighing above 400 kg, the MAE is slightly higher due to the limited data available for this weight range.

Another limitation is the controlled nature of the image data. The impact of challenging environmental conditions, such as low lighting, partial occlusion due to fences, or other visual obstructions has not been explored. These factors can significantly affect the quality of visual features.

Future research should address these limitations by incorporating more diverse datasets, conducting breed-specific evaluations, and assessing model robustness under varying environmental and image capturing conditions.

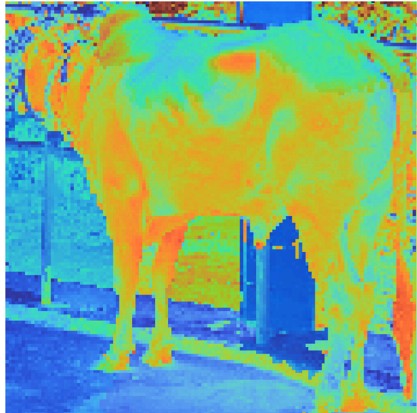 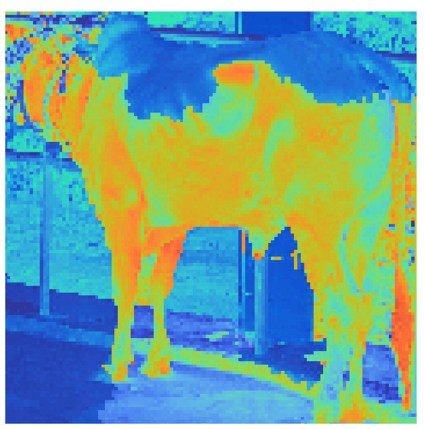

(a) Cattle's weight is predicted using 2Conv2Dense model, where the Actual Weight is : 252 kg, and Predicted Weight is : 234.33 kg.

b) Cattle's weight is predicted using 3Conv3Dense model, where the Actual Weight is : 252 kg, and Predicted Weight is : 253.66 kg.

**Fig 14**. **Error case analysis using LIME visualization on cattle's weight prediction.**

## 6 Conclusion and future work

In this study, we introduced CattleNet-XAI, a custom 3Conv3Dense CNN model built to estimate cattle weight from 2D images. The model uses a focused preprocessing process, including image-to-array conversion, normalization, and histogram equalization, along with a lightweight design suited for livestock images. Tested on the Cow Image Dataset (2,044 images from eight breeds), it outperformed other custom CNN models, the pre-trained EfficientNetB3 model, and traditional regression methods (Random Forest, Linear Regression), achieving an MAE of 18.02 kg, an RMSE of 19.85 kg, and an $R^2$ of 94.32%. This shows that a domain-specific, streamlined deep learning model can deliver high accuracy at low computational cost, making it suitable for practical farm use.

A key part of this work is using LIME explainability to show how the model makes its predictions. The results clearly highlighted the rib cage, abdomen, and hindquarters as the most influential regions for weight prediction, aligning closely with the knowledge of livestock experts. This level of transparency builds trust and encourages adoption in farming, where both accuracy and clarity are important. By combining high accuracy with clear explanations, CattleNet-XAI provides a scalable, non-invasive alternative to traditional weighing methods that can be slow and error-prone.

In the future, we intend to enhance the accuracy and applicability of the model by incorporating additional variables such as environmental factors and feed intake, which may improve weight estimation. A primary objective is to conduct a breed-specific performance analysis to evaluate the model's consistency across different cattle breeds. We also plan to develop a real-time weight estimation tool, which will involve exploring advanced data augmentation techniques and neural network architectures to ensure reliable performance under varying farm conditions. In addition, a breed-specific performance analysis will be performed to assess the consistency of the model between different breeds of cattle. These improvements aim to make the model more practical and easier to use on different types of farms.

## Acknowledgments

This research is supported by the Independent University, Bangladesh (IUB) Sponsored Research.

## Author contributions

**Conceptualization:** Md Junayed Hossain, Jannatul Ferdaus.

**Data curation:** Md Junayed Hossain.

**Formal analysis:** Md Junayed Hossain, Jannatul Ferdaus.

**Investigation:** Md Junayed Hossain, Jannatul Ferdaus.

**Methodology:** Md Junayed Hossain, Jannatul Ferdaus.

**Project administration:** Ashraful Islam, M. Ashraful Amin.

**Resources:** Ashraful Islam, M. Ashraful Amin.

**Supervision:** Ashraful Islam, M. Ashraful Amin.

**Validation:** Md Junayed Hossain, Jannatul Ferdaus, Ashraful Islam, M. Ashraful Amin.

**Visualization:** Md Junayed Hossain, Jannatul Ferdaus.

**Writing – original draft:** Md Junayed Hossain, Jannatul Ferdaus.

**Writing – review & editing:** Md Junayed Hossain, Jannatul Ferdaus, Ashraful Islam, M. Ashraful Amin.

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
