## [Decision Letter · Decision Letter 0]

21 Apr 2025

PONE-D-25-09685CattleNet-XAI: An Explainable CNN Framework for Efficient Cattle Weight EstimationPLOS ONE

Dear Dr. Islam,

Thank you for submitting your manuscript to PLOS ONE. After careful consideration, we feel that it has merit but does not fully meet PLOS ONE’s publication criteria as it currently stands. Therefore, we invite you to submit a revised version of the manuscript that addresses the points raised during the review process.

**ACADEMIC EDITOR: Major Revision**

We look forward to receiving your revised manuscript.

Kind regards,

Agbotiname Lucky Imoize

Academic Editor

PLOS ONE

Additional Editor Comments:

The authors should revise the paper according to the reviewers' comments and improve the English significantly.

Reviewers' comments:

Reviewer's Responses to Questions

**Comments to the Author**

1. Is the manuscript technically sound, and do the data support the conclusions?

Reviewer #1: Partly

Reviewer #2: Partly

2. Has the statistical analysis been performed appropriately and rigorously? 

Reviewer #1: No

Reviewer #2: Yes

3. Have the authors made all data underlying the findings in their manuscript fully available?

Reviewer #1: Yes

Reviewer #2: Yes

4. Is the manuscript presented in an intelligible fashion and written in standard English?

Reviewer #1: Yes

Reviewer #2: Yes

5. Review Comments to the Author

Reviewer #1: 1. Metrics

Table 2 compares MAE values with studies using much smaller datasets (e.g., 20–34 cattle). The authors should contextualize these comparisons, as their larger dataset (513 cattle) may inherently enable better performance.

How do the authors account for dataset size differences when comparing performance? Were additional metrics (e.g., MAPE) considered for a fairer comparison?

The manuscript compares MAE with what authors refer to as “recent previous” papers. I have two concerns here. Metrics like MAE or RMSE can be influenced by the scale of the data, and MAPE is often preferred when comparing predictive models across different studies. RMSE is the most commonly used indicator of predictive performance and I have no issues with using it to compare models using the same dataset. It’s the use of metrics like MAE and RMSE for comparison between studies that I have issues with. If a study used smaller cows on average (due to any reason such as the breed or the age of sampled cows, the farm situation, etc), it will surely result in lower RMSE a priori compared to a study that used on average larger cows. In addition, the comparisons need to include most recent publications as the research area is rapidly advancing. For instance, recent papers such as those of Gebreyesus et al. ( https://doi.org/10.3389/fgene.2022.947176 ) and Hou et al (https://doi.org/10.1016/j.compag.2023.108184 ), both from 2023, show lower MAPE values.

2. Prediction of body weight

The system implemented here might be considered an overkill. Body weight have been shown to be one of the easiest traits to predict from images given the high correlation between dimension (captured from images/pixel size) and body weight across species. Several studies show the possibility of BW prediction with simpler setups such as a single overhead camera and simpler models. The system proposed here makes sense only if additional value is taken out of it such as the prediction of other, more challenging conditions like lameness or stress responses. Given the authors claim to address the need for high throughput and easier measuring system, why go for complex models?

3. High-throughput?

The authors claim that their method addresses the need for high-throughput cattle weight estimation. High-throughput applications in real-world farm settings involve multiple animals, occlusions, and complex environments (e.g., fences, farm equipment, varied lighting conditions). he dataset used in this study appears to consist of controlled images without clear discussion of occlusions or real-world production settings.

Were images captured in open farm environments, or were they taken under controlled conditions?

Does the model account for partial occlusions (e.g., cattle overlapping, fences blocking parts of the body)?

How does the model handle low-resolution or motion-blurred images, which are common in farm settings?

4. Novelty & justification of approach

The study introduces CNN-based cattle weight estimation, but similar approaches have been explored (e.g., EfficientNet-based livestock weight prediction, Transformer-based vision models).

What specific advancements does CattleNet-XAI offer over prior deep learning models? A more in-depth comparison with state-of-the-art methods (not just RF and LR) is needed.

5. Model interpretability (explainable AI - XAI)

The paper claims to use LIME (Local Interpretable Model-agnostic Explanations) for explainability, but there is insufficient analysis.

What were the most influential features in LIME? Were certain image regions consistently emphasized in weight predictions?

Could SHAP (SHapley Additive Explanations) be a better alternative to LIME in this case?

6. Dataset bias and generalizability

The dataset contains 17,899 images, but only 2052 are original images. The impact of synthetic images or augmentations on model performance and bias needs discussion.

How diverse are the cattle breeds, and do model errors differ by breed, age, or body size?

Can this model generalize to different environments, lighting conditions, or angles?

The dataset includes 513 unique cattle with multiple images per individual. Without clear details on how the data were split (e.g., ensuring no overlap of the same cattle in training and test sets), there is a high risk of data leakage, which could artificially inflate model performance. This is a major scientific flaw that undermines the validity of the results.

How were the images of the same cattle partitioned across training, validation, and test sets? Were subject-wise splits used to ensure independence?

The authors apply normalization before histogram equalization, which is unconventional. Typically, histogram equalization is performed first to enhance contrast, followed by normalization. This sequence could affect feature extraction and model performance.

Question: Why was this order chosen? Was there empirical evidence supporting this approach? If not, consider revising the pipeline.

The dataset is heavily skewed toward the LOCAL breed (68.8%), with underrepresented breeds like BRAHMA and MIR KADIM (0.4% each). This raises concerns about the model's generalizability to other breeds.

Were techniques like stratified sampling, data augmentation, or weighted loss functions used to address class imbalance? If not, how do the authors justify the model's applicability to underrepresented breeds?

7. Comparison with advanced architectures

The study primarily compares the proposed CNN model to Random Forest and Linear Regression, which are known to be weaker for image-based tasks.

How would CattleNet-XAI compare against pretrained vision models like ResNet, EfficientNet, or Vision Transformers (ViTs)?

Were pre-trained networks (transfer learning) considered? If not, why?

The authors use YOLOv5 for feature extraction but do not justify why this version was chosen over newer alternatives (e.g., YOLOv8) or other object detection models. Additionally, the number of features selected via Recursive Feature Elimination (RFE) and their interpretability are not discussed.

Question: Why was YOLOv5 chosen? How many features were selected via RFE, and what were the most important features for weight prediction?

8. Hyperparameter tuning & optimization

The paper does not describe how hyperparameters (e.g., learning rate, batch size, number of filters, dropout rates) were selected.

Was any hyperparameter tuning done (e.g., grid search, Bayesian optimization)?

Without tuning, the performance gains of the CNN model may not be fully optimized.

The authors propose a 3Conv3Dense CNN but do not provide a thorough justification for this architecture. Why were three convolutional layers and three dense layers chosen? Were other architectures (e.g., ResNet, EfficientNet) explored?

9. Lack of Statistical significance testing

Model performance is reported with MAE, MSE, RMSE, and R², but confidence intervals or statistical significance tests are missing.

Were differences between models statistically significant? Could paired t-tests or Wilcoxon signed-rank tests confirm these differences?

10. Overfitting and generalization issues

The validation loss curves indicate some degree of overfitting, especially in shallower CNN architectures.

Was dropout or regularization (L1/L2) used to mitigate overfitting?

Did the model undergo k-fold cross-validation, or was only a single train-test split used?

The paper lacks details on cross-validation, train-test split ratios, and hyperparameter tuning. Without this information, reproducibility is compromised.

What were the train-test split ratios? Were hyperparameters tuned using cross-validation? If so, what were the optimal values?

11. Regression model feature engineering

The Recursive Feature Elimination (RFE) process is mentioned but lacks sufficient detail.

What features were selected as most important?

Would feature extraction via Principal Component Analysis (PCA) improve the performance of regression models?

Reviewer #2: This study presents a CNN built for estimating cattle weight. The results show a relevant improvement in the method's accuracy compared to the traditional random forest and linear regression methods. The study is relevant since it offers immediately applicable advantages.

Comments:

The writing is very clear and grammatically correct. Different methods are presented educationally, thus, this paper could serve as a quick reference tool for the field of traditional machine learning algorithms.

At some passages, however, the writing seems exaggerated. For example, the last line in the Abstract says “presenting a revolutionary method for cattle management”. In the conclusions, it reads “The model’s exceptional performance underscores the promise of deep learning in automating agricultural activities, diminishing dependence on manual and error-prone weight estimation techniques. This innovation could improve precision agriculture by providing a scalable, non-invasive method for real-time livestock management.” This excessive flourish tone is not coherent with an objective interpretation of the results of an experiment or the evaluation of a method. I think the inappropriateness of this style may not reflect the authors’ spirit about their work. Perhaps this results from using an automatic writing polishing tool, which tends to push beyond convenient limits, the impact the text will create. This is only my explanation, and it may well be pure speculation. But the fact is that this paper’s content compares several techniques that existed dozens of years ago.

Presenting figures next to their caption makes reviewing this manuscript difficult. Please put the Figures in their place.

6. PLOS authors have the option to publish the peer review history of their article (what does this mean?). If published, this will include your full peer review and any attached files.

Reviewer #1: No

Reviewer #2: **Yes: **Gerardo Febres

---

## [Author Response · Author response to Decision Letter 1]

4 Jun 2025

*** Response to Reviewers file is also attached as a PDF during the file submission.***

Response to Reviewers

Manuscript Number: PONE-D-25-09685

Title: CattleNet-XAI: An Explainable CNN Frame- work for Efficient Cattle Weight Estimation

The authors would like to thank the editor and reviewers for their constructive comments and suggestions that have helped improve the quality of this manuscript. The manuscript has undergone a thorough revision according to the reviewers’ comments. Please see our responses below (Pages 1–6 for Reviewer-1 and Page 7 for Reviewer-2). For the reviewers’ convenience, we have made significant changes in the revised manuscript and highlighted in Yellow.

Reviewer 1

Reviewer Point #1 — Table 2 compares MAE values with studies using much smaller datasets (e.g., 20–34 cattle). The authors should contextualize these comparisons, as their larger dataset (513 cattle) may inherently enable better performance. How do the author’s account for dataset size differences when comparing performance? Were additional metrics (e.g., MAPE) considered for a fairer comparison? The manuscript compares MAE with what authors refer to as “recent previous” papers. I have two concerns here. Metrics like MAE or RMSE can be influenced by the scale of the data, and MAPE is often preferred when comparing predictive models across different studies. RMSE is the most commonly used indicator of predictive performance and I have no issues with using it to compare models using the same dataset. It’s the use of metrics like MAE and RMSE for comparison between studies that I have issues with. If a study used smaller cows on average (due to any reason such as the breed or the age of sampled cows, the farm situation, etc), it will surely result in lower RMSE a priori compared to a study that used on average larger cows. In addition, the comparisons need to include most recent publications as the research area is rapidly advancing. For instance, recent papers such as those of Gebreyesus et al. ( https://doi.org/10.3389/fgene.2022.947176 ) and Hou et al (https://doi.org/10.1016/j.compag.2023.108184 ), both from 2023, show lower MAPE values.

Reply: We thank the reviewer for highlighting this important point. In response, we have revised the manuscript to improve the fairness and clarity of our cross-study comparisons. Inclusion of MAPE for Fairer Cross-Study Comparisons: To ensure that our comparisons are robust across studies involving cattle of varying breeds, sizes, and weight distributions, we have included Mean Absolute Percentage Error (MAPE) as an additional evaluation metric. Unlike MAE and RMSE, which are sensitive to scale, MAPE provides a normalized, scaleindependent measure of error. This allows for more meaningful performance comparisons between studies that may involve different weight ranges or cattle types. The revised (Table 1) now includes MAPE values for all models, and we have updated the Results section accordingly to reflect and interpret these values in line 648. It is observed that the 3Conv3Dense model achieved a MAPE of approximately 6.22%.

Reviewer Point #2 — Prediction of body weight The system implemented here might be considered an overkill. Body weight have been shown to be one of the easiest traits to predict from images given the high correlation between dimension (captured from images/pixel size) and body weight across species. Several studies show the possibility of BW prediction with simpler setups such as a single overhead camera and simpler models. The system proposed here makes sense only if additional value is taken out of it such as the prediction of other, more challenging conditions like lameness or stress responses. Given the authors claim to address the need for high throughput and easier measuring system, why go for complex models?

Reply: Thank you for the valuable comment. We agree that body weight is often considered one of the easier traits to predict due to its high correlation with body dimensions. However, our study focused on using real-world 2D cattle images. In such cases, using a CNN-based approach allows the model to learn complex visual patterns beyond basic height and width. While simpler models like linear regression and random forest performed reasonably well, our results clearly show that our custom CNN architecture (3Conv3Dense) significantly outperformed them as well as the more complex pre-trained EfficientNetB3, which is implemented in our revised version. For example, our CNN achieved a lower MAE (18.02 kg) compared to EfficientNetB3 (21.05 kg), despite being lighter and more tailored to our specific task. Additionally, we used LIME to explain CNN’s predictions, which highlighted regions like the rib cage and abdomen as key contributors to the model’s decisions. This explainability is crucial for trust and transparency in real-world use. Our model offers a balanced approach as it is more robust than basic regressors, simpler than heavy models like EfficientNetB3, and still efficient enough for practical deployment. While predicting weight was our main goal, the proposed system is designed to be scalable for more complex traits like lameness detection in the future.

Reviewer Point #3 — High-throughput? The authors claim that their method addresses the need for high-throughput cattle weight estimation. High-throughput applications in real-world farm settings involve multiple animals, occlusions, and complex environments (e.g., fences, farm equipment, varied lighting conditions). he dataset used in this study appears to consist of controlled images without clear discussion of occlusions or real-world production settings.

Were images captured in open farm environments, or were they taken under controlled conditions?

Does the model account for partial occlusions (e.g., cattle overlapping, fences blocking parts of the body)?

How does the model handle low-resolution or motion-blurred images, which are common in farm settings?

Reply: We thank the reviewer for raising this important point regarding the real-world applicability of our system in high-throughput farm environments. Our current dataset consists primarily of side-view images of individual cattle, many of which were taken in relatively controlled settings with minimal occlusion. We acknowledge that this does not fully represent the complexity of real-world farm scenarios, where animals may overlap, lighting may vary, and motion blur or background clutter can impact image quality. At this stage, our primary goal was to develop a proof-of-concept model that could accurately predict cattle weight using unconstrained, side-view images without requiring physical measurements or fixed-camera setups. However, we have incorporated histogram equalization and normalization to improve contrast and robustness to lighting variability. In future work, we plan to expand the dataset to include images from operational farms, featuring occlusions, multiple animals, varying lighting, and different angles. We have now revised the manuscript to clearly state the limitations of our current dataset and added a discussion on our future plans for real-world deployment.

Reviewer Point #4 — Novelty & justification of approach The study introduces CNN-based cattle weight estimation, but similar approaches have been explored (e.g., EfficientNet-based livestock weight prediction, Transformer-based vision models). What specific advancements does CattleNet-XAI offer over prior deep learning models? A more in-depth comparison with state-of-the-art methods (not just RF and LR) is needed.

Reply: Thank you for the valuable feedback. Our primary contribution lies in integrating LIME-based interpretability with a custom CNN model, which we refer to as CattleNet-XAI. This combination provides not only accurate cattle weight predictions but also model transparency which enables insights into which regions of the images influence predictions, which is critical for trust and adoption in agricultural sectors. While prior works have explored deep learning models like EfficientNet and transformer-based architectures for livestock analysis, few have emphasized explainability tailored to end-user needs. In response to the reviewer’s suggestion, we have additionally implemented the EfficientNetB3 model highlighted in line number 469 in our revised manuscript and incorporated its results into our comparative analysis [Table-1]. This allows for a deeper evaluation of CattleNet-XAI against a state-of-the-art baseline, further validating its performance and practical relevance.

Reviewer Point #5 — Model interpretability (explainable AI - XAI) The paper claims to use LIME (Local Interpretable Model-agnostic Explanations) for explainability, but there is insufficient analysis. What were the most influential features in LIME? Were certain image regions consistently emphasized in weight predictions? Could SHAP (SHapley Additive Explanations) be a better alternative to LIME in this case?

Reply: We appreciate the reviewer’s insightful question regarding model interpretability. LIME was used to visualize which parts of the image contributed most to the predicted weight. Our analysis showed that LIME consistently highlighted regions such as the rib cage, hindquarters, and abdomen areas that are logically correlated with muscle and fat mass, thus aligning well with visual weight assessments used by livestock experts. We agree that SHAP is a powerful alternative for explainability, particularly in structured/tabular data or when model internals are accessible. However, since our pipeline involves CNNs trained on raw image inputs, LIME was chosen for its flexibility in image-based local interpretability. Nonetheless, in future work, we plan to incorporate Deep SHAP or Grad-CAM to provide complementary global explanations at the feature map level. We also update our revised manuscript in that manner in line number 859.

Reviewer Point #6 — The dataset contains 17,899 images, but only 2052 are original images. The impact of synthetic images or augmentations on model performance and bias needs discussion. How diverse are the cattle breeds, and do model errors differ by breed, age, or body size? Can this model generalize to different environments, lighting conditions, or angles? The dataset includes 513 unique cattle with multiple images per individual. Without clear details on how the data were split (e.g., ensuring no overlap of the same cattle in training and test sets), there is a high risk of data leakage, which could artificially inflate model performance. This is a major scientific flaw that undermines the validity of the results. How were the images of the same cattle partitioned across training, validation, and test sets? Were subject-wise splits used to ensure independence? The authors apply normalization before histogram equalization, which is unconventional. Typically, histogram equalization is performed first to enhance contrast, followed by normalization. This sequence could affect feature extraction and model performance. Question: Why was this order chosen? Was there empirical evidence supporting this approach? If not, consider revising the pipeline. The dataset is heavily skewed toward the LOCAL breed (68.8%), with underrepresented breeds like BRAHMA and MIR KADIM (0.4% each). This raises concerns about the model’s generalizability to other breeds. Were techniques like stratified sampling, data augmentation, or weighted loss functions used to address class imbalance? If not, how do the authors justify the model’s applicability to underrepresented breeds?

Reply: In this study, we applied image normalization prior to histogram equalization, deviating from the conventional order. Specifically, pixel values were first scaled to the [0, 1] range to standardize the intensity distribution across images. This step reduces the impact of lighting variability and prepares the data for consistent histogram operations. After normalization, the image was converted back to an 8-bit format to enable channel-wise histogram equalization, enhancing local contrast in a controlled manner. Finally, the image was re-normalized to [0, 1] for input into the CNN. This ordering was found to improve training stability and validation performance, particularly by preventing over-enhancement in underexposed regions. Although the dataset contains an imbalanced distribution of cattle breeds, we did not apply oversampling techniques, as the model’s predictions are primarily based on body features such as the rib cage, abdomen, and overall body shape characteristics and that remain largely consistent across different breeds. While there are some breed-specific differences in traits like color, head shape, or horn structure, as discussed in our XAI section (line 859), these factors have minimal influence on weight estimation in our model.

We may analyze breed-wise prediction accuracy. In the future, we will evaluate the model’s performance by breed to improve and validate our approach.

Reviewer Point #7 — Comparison with advanced architectures The study primarily compares the proposed CNN model to Random Forest and Linear Regression, which are known to be weaker for image-based tasks. How would CattleNet-XAI compare against pretrained vision models like ResNet, EfficientNet, or Vision Transformers (ViTs)? Were pre-trained networks (transfer learning) considered? If not, why? The authors use YOLOv5 for feature extraction but do not justify why this version was chosen over newer alternatives (e.g., YOLOv8) or other object detection models. Additionally, the number of features selected via Recursive Feature Elimination (RFE) and their interpretability are not discussed. Question: Why was YOLOv5 chosen? How many features were selected via RFE, and what were the most important features for weight prediction?

Reply: In response to your suggestions, we expanded our study by implementing the EfficientNetB3 model and including its results in our comparative analysis. Deatils are descrived in line number 469. also it’s results with other graphs such as validation curves, predicted graph or residual graphs are also discussed in each section.

We selected YOLOv5 for object-centric feature extraction due to its strong trade-off between speed and accuracy, along with ease of integration. Although newer versions like YOLOv8 are available, YOLOv5 was effective and stable at the time of our experiments. All the reasoning are discussed in our revised menuscript from line number 313 in our Feature Extraction using YOLOv5 section.

RFE was used to select the most predictive features from the high-dimensional CNN feature space. While RFE helps reduce model complexity and improve accuracy, it does not directly offer anatomical interpretability. To address this, we employed LIME-based visual explanations, which consistently highlighted regions such as the rib cage, abdomen, and hindquarters as contributing most to the model’s predictions. This combination of feature selection and visual interpretation confirms that our model leverages semantically meaningful features for cattle weight estimation. we have included everything in our revised manuscript in line number 341.

Reviewer Point #8 — Hyperparameter tuning optimization

The paper does not describe how hyperparameters (e.g., learning rate, batch size, number of filters, dropout rates) were selected. Was any hyperparameter tuning done (e.g., grid search, Bayesian optimization)? Without tuning, the performance gains of the CNN model may not be fully optimized.

The authors propose a 3Conv3Dense CNN but do not provide a thorough justification for this architecture. Why were three convolutional layers and three dense layers chosen? Were other architectures (e.g., ResNet, EfficientNet) explored?

Reply: Thank You for your comment. We have revised the manuscript to provide a more detailed explanation of our hyperparameter tuning and architectural decisions (Section 3.4: Hyperparameter Tuning and Optimization in line number 532). Specifically, we performed empirical tuning of key hyperparameters, including learning rate, batch size, dropout rate, and L2 regularization. While we did not use automated search techniques such as grid search or Bayesian optim

---

## [Decision Letter · Decision Letter 1]

1 Aug 2025

PONE-D-25-09685R1CattleNet-XAI: An Explainable CNN Framework for Efficient Cattle Weight EstimationPLOS ONE

Dear Dr. Islam,

Thank you for submitting your manuscript to PLOS ONE. After careful consideration, we feel that it has merit but does not fully meet PLOS ONE’s publication criteria as it currently stands. Therefore, we invite you to submit a revised version of the manuscript that addresses the points raised during the review process.

**ACADEMIC EDITOR: Major revision**==============================

We look forward to receiving your revised manuscript.

Kind regards,

Agbotiname Lucky Imoize

Academic Editor

PLOS ONE

Journal Requirements:

Additional Editor Comments:

Dear Authors,

Please revise your paper according to the reviewers' comments.

Note that reviewer 3 has suggested their work to be cited in this paper. Authors must not cite unrelated references in the current work.

Thank you.

Reviewers' comments:

Reviewer's Responses to Questions

**Comments to the Author**

1. If the authors have adequately addressed your comments raised in a previous round of review and you feel that this manuscript is now acceptable for publication, you may indicate that here to bypass the “Comments to the Author” section, enter your conflict of interest statement in the “Confidential to Editor” section, and submit your "Accept" recommendation.

Reviewer #2: (No Response)

Reviewer #3: All comments have been addressed

2. Is the manuscript technically sound, and do the data support the conclusions?

Reviewer #2: No

Reviewer #3: Partly

3. Has the statistical analysis been performed appropriately and rigorously? 

Reviewer #2: Yes

Reviewer #3: No

4. Have the authors made all data underlying the findings in their manuscript fully available?

Reviewer #2: (No Response)

Reviewer #3: No

5. Is the manuscript presented in an intelligible fashion and written in standard English?

Reviewer #2: (No Response)

Reviewer #3: No

6. Review Comments to the Author

Reviewer #2: Some of my comments in the former review were addressed. No doubt the manuscript has improved; The writing is clean, and the structure of the document, together with the completeness of the types of neural networks included in the tests, makes this study a valid consultation document. However, I feel a lack of novelty in the study that could be considered conflicting with some assertions made at the end of the document. In the conclusions, for example, the document says:

“The model’s exceptional performance underscores the promise of deep learning in automating agricultural activities, diminishing dependence on manual and error-prone weight estimation techniques. This innovation could improve precision agriculture by providing a scalable, non-invasive method for real-time livestock management.”

When the reader intends to identify which is the model that has EXCEPTIONAL performance, the reader realizes this is a difficult task. Perhaps that is because the subject of that exceptionality does not exist in the document. At least I do not see a model proposed in the study. Finally, the Section Conclusions do not show a conclusion. Instead, a vague text rich in rhetoric that, in my opinion, does not fit well in the conclusions segment of a scientific publication.

Reviewer #3: 1. The paper lacks in major contribution and motivation of research.

2. The background and significance of this study should be highlighted in the abstract.

3. Check the English presentation of this paper to remove the typo mistakes. Some grammatical issues need to be addressed in the whole text. Please reform the long paragraphs. Please polish the writing and English of the manuscript carefully. The writing of the paper needs a lot of improvement in terms of grammar, spelling, and presentation. The paper needs careful English polishing since there are many typos and poorly written sentences. I found several errors.

4. In the "Introduction" section, a more detailed analysis of the existing literature on the subject is needed, and an in-depth analysis of the possible application fields.

5. The mathematics used throughout the article is still not very strict. Please try to update and illustrate some elements in the mathematical model that are not defined very strictly.

6. The overall structure of the article should be improved.

7. The result part is week, results and discussion should be better explained.

8. References must be updated and add the suitable from the following

https://doi.org/10.3390/en12050961, https://doi.org/10.1109/EIConRus.2018.8317170, https://doi.org/10.1016/j.est.2024.113556, https://doi.org/10.1109/ACCESS.2024.3437191, https://doi.org/10.1109/MEPCON.2017.8301313, https://dx.doi.org/10.21608/jaet.2021.82231, https://doi.org/10.1109/MEPCON47431.2019.9008171, https://doi.org/10.21608/sej.2021.155557,

https://doi.org/10.1109/MEPCON58725.2023.10462371, DOI: 10.1109/ACCESS.2024.3525183, https://doi.org/10.1007/s00521-024-09433-3, https://doi.org/10.1371/journal.pone.0317619, https://doi.org/10.20508/ijrer.v14i2.14346.g8898, https://doi.org/10.20508/ijrer.v13i1.13718.g8659, https://doi.org/10.1007/s00521-024-09433-3, https://doi.org/10.1007/s00521-024-09902-9, https://doi.org/10.21608/SVUSRC.2024.279389.1198,

9. Check all of your Figures and Tables have a good explanation of your text.

10. Many paragraphs without citations

11. What are the contributions and novelty of work mentioned?

12. The authors' conclusions need to be improved, a comparison of the results obtained with those already existing in the literature would be appropriate. I suggest also describing what can still be improved in this work, which can still be improved based on the results obtained, according to the authors' view. It is suggested to offer some limitations existed in this study and an outlook for future study in the last section.

7. PLOS authors have the option to publish the peer review history of their article (what does this mean?). If published, this will include your full peer review and any attached files.

Reviewer #2: No

Reviewer #3: No

---

## [Author Response · Author response to Decision Letter 2]

23 Aug 2025

A separate file for the Response to Reviewers is attached as PDF file for your convenience.

Response to Reviewers

Manuscript Number: PONE-D-25-09685R2

Title: CattleNet-XAI: An Explainable CNN Framework for Efficient Cattle Weight Estimation

The authors would like to thank the editor and reviewers for their constructive comments

and suggestions that have helped improve the quality of this manuscript. The manuscript

has undergone a thorough revision according to the reviewers’ comments. Please see our

responses below. For the reviewers’

convenience, we have made significant changes in the revised manuscript and highlighted

in Yellow.

Reviewer 1

No comment was found from Reviewer 1 during this review feedback from the journal.

Reviewer 2

Reviewer Point #1 —

Some of my comments in the former review were addressed. No doubt the manuscript

has improved; The writing is clean, and the structure of the document, together with the

completeness of the types of neural networks included in the tests, makes this study a

valid consultation document. However, I feel a lack of novelty in the study that could

be considered conflicting with some assertions made at the end of the document. In the

conclusions, for example, the document says:

“The model’s exceptional performance underscores the promise of deep learning in

automating agricultural activities, diminishing dependence on manual and error-prone

weight estimation techniques. This innovation could improve precision agriculture by

providing a scalable, non-invasive method for real-time livestock management.”

When the reader intends to identify which is the model that has EXCEPTIONAL

performance, the reader realizes this is a difficult task. Perhaps that is because the

subject of that exceptionality does not exist in the document. At least I do not see a model

proposed in the study. Finally, the Section Conclusions do not show a conclusion. Instead,

a vague text rich in rhetoric that, in my opinion, does not fit well in the conclusions

segment of a scientific publication.

Reply: Thank you for your valuable feedback. We have made updates based on your suggestions,

including rewriting the entire Conclusion and Future Work sections (in line number

920). Additionally, we have highlighted our novelty in both the abstract and introduction

sections in line number 67.

Reviewer 3

Reviewer Point #1 —

The paper lacks in major contribution and motivation of research.

Reply:

We thank the reviewer for their feedback. Our research aims to solve the main problems

with how cattle weight is currently measured. Existing methods are either too expensive, like

3D systems , or they are “black box” AI models that farmers don’t trust because it’s unclear

how they work. Our major contribution is CattleNet-XAI framework, which presents a novel,

cost-effective solution using 2D images. This framework features a custom “3Conv3Dense”

CNN that achieves a state-of-the-art Mean Absolute Error of 18.02 kg, significantly outperforming

prior work. Critically, its novelty lies in the integration of Explainable AI (XAI) using

LIME, which provides visual proof that the model bases its predictions on anatomically relevant

features like the rib cage and abdomen. By being both highly accurate and transparent, our

method helps build the trust needed for AI in agriculture, which is a significant and practicle.

And all of these contributions are explicitly listed at the end of the Introduction (Line no 67)

in our manuscript.

Reviewer Point #2 —

The background and significance of this study should be highlighted in the abstract.

Reply: Thank you for your comment. We have revised the abstract according to your

comment. We have added the background of the study and clearly explained its significance

in our revised abstract.

Reviewer Point #3 — Check the English presentation of this paper to remove the

typo mistakes. Some grammatical issues need to be addressed in the whole text. Please

reform the long paragraphs. Please polish the writing and English of the manuscript

carefully. The writing of the paper needs a lot of improvement in terms of grammar,

spelling, and presentation. The paper needs careful English polishing since there are

many typos and poorly written sentences. I found several errors.

Reply: We sincerely thank the reviewer for their valuable feedback. In response, we have

gone through the entire text to correct all grammatical errors and typos (like in line number

80, 109, 140, 192, 241, 284 etc.). We have also restructured sentences and paragraphs to

improve clarity, flow, and overall readability. And all the changes are marked as yellow in our

revised manuscript.

Reviewer Point #4 —

In the “Introduction” section, a more detailed analysis of the existing literature on

the subject is needed, and an in-depth analysis of the possible application fields.

Reply:

The entire introduction has been carefully reviewed. Although we have a separate literature

review section still based on your feedback, we have updated the introduction section

accordingly and added a short paragraph (lines 28–42) discussing existing work on this subject

along with its potential applications.

Reviewer Point #5 —

The mathematics used throughout the article is still not very strict. Please try to

update and illustrate some elements in the mathematical model that are not defined very

strictly.

Reply: We have gone through each of the mathematics updated where needed also we have

added three new equations for Dropout Rate (in line number 554), L2 Regularization (in line

number 557) and R2 value (in line number 639).

Reviewer Point #6 —

The overall structure of the article should be improved.

Reply: We appreciate the reviewer’s suggestion to improve the article’s structure. In response,

we have carefully revised the manuscript to enhance this. We focused on strengthening

the narrative in the introduction as well as conclusion. Also improved the transitions

between sections, and better organizing the Result Analysis section with clearer subheadings

to guide the reader. We believe these revisions have made the paper’s structure much stronger

and easier to follow.

Reviewer Point #7 —

The result part is week, results and discussion should be better explained.

Reply: We sincerely thank the reviewer for this constructive feedback. We agree that the

Results and Discussion section required more depth and have revised it extensively to better

explain our findings.

The updated section (from line number: 641 ) now provides a more detailed description

of what each performance metric signifies in a practical context. We have also expanded the

comparative analysis to better explain why our proposed model outperformed the baselines,

discussing the impact of architectural choices and the advantages of a domain-specific model.

Reviewer Point #8 —

References must be updated and add the suitable from the following

https://doi.org/10.3390/en12050961, https://doi.org/10.1109/EIConRus.2018.8317170,

https://doi.org/10.1016/j.est.2024.113556, https://doi.org/10.1109/ACCESS.2024.3437191,

https://doi.org/10.1109/MEPCON.2017.8301313, https://dx.doi.org/10.21608/jaet.2021.82231,

https://doi.org/10.1109/MEPCON47431.2019.9008171, https://doi.org/10.21608/sej.2021.155557,

https://doi.org/10.1109/MEPCON58725.2023.10462371, DOI: 10.1109/ACCESS.2024.3525183,

https://doi.org/10.1007/s00521-024-09433-3, https://doi.org/10.1371/journal.pone.0317619,

https://doi.org/10.20508/ijrer.v14i2.14346.g8898,

https://doi.org/10.20508/ijrer.v13i1.13718.g8659, https://doi.org/10.1007/s00521-024-09433-

3, https://doi.org/10.1007/s00521-024-09902-9,

https://doi.org/10.21608/SVUSRC.2024.279389.1198,

Reply:

Thank you for your suggestions. We found that two of the suggested papers are relevant

to our work and have cited them at lines 526 and 630.

Reviewer Point #9 —

Check all of your Figures and Tables have a good explanation of your text.

Reply: Thank you for your comment. We have checked and updated extensively where

needed like we have updated Fig-6 caption in line number 389. Also Figure -13 caption in

line number 393.

Reviewer Point #10 —

Many paragraphs without citations

Reply: We have carefully reviewed the entire article and added citations where necessary to

better support our claims and methods.

Reviewer Point #11 —

What are the contributions and novelty of work mentioned?

Reply: The primary contribution and novelty of this research are consolidated in CattleNet-

XAI, a framework that introduces a high-performance, explainable, and rigorously validated

solution for cattle weight estimation. The core of this work is the development of a custom

3Conv3Dense CNN architecture, which is a novel, lightweight model tailored specifically for

this weight prediction task.

We address the critical “black box” issue in agricultural AI by integrating Explainable AI

(XAI) using LIME visualizations. This provides transparent, trustworthy evidence that the

model’s predictions are based on anatomically correct regions like the rib cage and abdomen,

enhancing its practical value. All of this is highlighted in the introduction section of our

manuscript, specifically at line 67.

Reviewer Point #12 —

The authors’ conclusions need to be improved, a comparison of the results obtained

with those already existing in the literature would be appropriate. I suggest also describing

what can still be improved in this work, which can still be improved based on the

results obtained, according to the authors’ view. It is suggested to offer some limitations

existed in this study and an outlook for future study in the last section.

Reply:

Thank you for your comment. We have thoroughly rewritten the entire Conclusion and

Future Work sections. Additionally, we already included a Limitations (Line 885) section

where we described the constraints of this study. A comparison table (Table 2, Line 840) is

also provided to contrast our work with existing studies.

---

## [Decision Letter · Decision Letter 2]

18 Sep 2025

PONE-D-25-09685R2CattleNet-XAI: An Explainable CNN Framework for Efficient Cattle Weight EstimationPLOS ONE

Dear Dr. Islam,

Thank you for submitting your manuscript to PLOS ONE. After careful consideration, we feel that it has merit but does not fully meet PLOS ONE’s publication criteria as it currently stands. Therefore, we invite you to submit a revised version of the manuscript that addresses the points raised during the review process.

**ACADEMIC EDITOR: Major Revision**==============================

We look forward to receiving your revised manuscript.

Kind regards,

Agbotiname Lucky Imoize

Academic Editor

PLOS ONE

Journal Requirements:

Additional Editor Comments:

Reviewer #4: Revise the paper according to the reviewers comments, and improve the English accordingly.

Reviewers' comments:

Reviewer's Responses to Questions

**Comments to the Author**

1. If the authors have adequately addressed your comments raised in a previous round of review and you feel that this manuscript is now acceptable for publication, you may indicate that here to bypass the “Comments to the Author” section, enter your conflict of interest statement in the “Confidential to Editor” section, and submit your "Accept" recommendation.

Reviewer #2: All comments have been addressed

Reviewer #4: All comments have been addressed

2. Is the manuscript technically sound, and do the data support the conclusions?

Reviewer #2: Yes

Reviewer #4: Partly

3. Has the statistical analysis been performed appropriately and rigorously? 

Reviewer #2: Yes

Reviewer #4: I Don't Know

4. Have the authors made all data underlying the findings in their manuscript fully available?

Reviewer #2: Yes

Reviewer #4: Yes

5. Is the manuscript presented in an intelligible fashion and written in standard English?

Reviewer #2: Yes

Reviewer #4: No

6. Review Comments to the Author

Reviewer #2: 2nd review

During the first and second reviews of this paper, I presented numerous criticisms about this study. The subject of the study is well supported and shows a valid aspect of reality deserving the application of CNN Models, thus, most of the problems presented in my reviews could be qualified as formal issues.

The authors have addressed the points I referred to in previous inspections. Including my concern about the lack of supported results of the study. In its current condition, the manuscript presents conclusions that are consistent with the quantitative results sustained in the text. The statistical assessment of the method is now well presented, and the formal aspects of mathematics is clearly shown in the text. In my opinion, this manuscript may now be published.

Reviewer #4: After reviewing the manuscript titled "CattleNet-XAI: An Explainable CNN Framework for Efficient Cattle Weight Estimation," I regret to inform that I recommend the rejection of this paper in its current form.

1. Lack of Novelty: The manuscript claims to present a novel framework, but upon review, the methodology closely follows existing techniques without introducing significant advancements. While the use of CNNs and XAI (Explainable AI) in livestock weight estimation is a relevant and growing field, the paper does not sufficiently differentiate itself from prior works. The proposed CNN model, though effective, lacks a clear distinction from other models already well-established in the literature. The contribution, as described, does not present sufficient novelty to justify publication.

2. Methodological Weakness: The explanation of the methodology is not sufficiently rigorous. For instance, the handling of the dataset, feature extraction techniques, and model performance evaluation need clearer justifications and explanations. The manuscript mentions advanced techniques like LIME for model interpretability but does not provide a comprehensive evaluation of how these methods are integrated and their actual impact on improving model transparency.

3. Unclear Results and Evaluation: While the paper presents results for the proposed models, the analysis is superficial. The comparison between different models (CNNs, EfficientNetB3, and traditional methods like Random Forest) is not sufficiently detailed, especially in terms of real-world applicability. A more thorough discussion on the limitations of the current model and areas for further improvement is needed.

4. Lack of Clear Conclusion: The conclusions drawn from the results are vague and lack specificity. The statement that the proposed model "could improve precision agriculture" seems speculative and not supported by a strong, comparative analysis of existing literature and methodologies. The absence of concrete future directions or a critical evaluation of the model's shortcomings makes the conclusion less impactful.

5. Writing and Grammar Issues: While some revisions were made, the manuscript still contains numerous grammatical errors and poorly constructed sentences. Several sections remain unclear, and the overall readability of the paper is compromised. A more thorough proofreading and restructuring are necessary.

In conclusion, while the topic is of potential interest, the manuscript lacks sufficient novelty, rigorous methodology, and a thorough analysis of results to merit publication. I recommend rejecting the paper and suggesting that the authors reconsider their approach, including refining the model's uniqueness, addressing the methodological gaps, and providing a more robust discussion of their findings.

7. PLOS authors have the option to publish the peer review history of their article (what does this mean?). If published, this will include your full peer review and any attached files.

Reviewer #2: **Yes: **Gerardo L. Febres

Reviewer #4: No

---

## [Author Response · Author response to Decision Letter 3]

22 Sep 2025

The authors would like to thank the editor and reviewers for their constructive comments

and suggestions that have helped improve the quality of this manuscript. The manuscript

has undergone a thorough revision according to the reviewers’ comments. Please see our

responses below (Pages 1 for Reviewer-2 and Page 2-3 for Reviewer-4). For the reviewers’

convenience, we have made significant changes in the revised manuscript and highlighted

in Yellow.

Reviewer 1

During this revision, no comment from Reviewer-1.

Reviewer 2

Reviewer Point # 1 —

During the first and second reviews of this paper, I presented numerous criticisms

about this study. The subject of the study is well supported and shows a valid aspect of

reality deserving the application of CNN Models, thus, most of the problems presented

in my reviews could be qualified as formal issues.

The authors have addressed the points I referred to in previous inspections. Including

my concern about the lack of supported results of the study. In its current condition,

the manuscript presents conclusions that are consistent with the quantitative results

sustained in the text. The statistical assessment of the method is now well presented,

and the formal aspects of mathematics is clearly shown in the text. In my opinion, this

manuscript may now be published.

Reply: Thank you for your valuable feedback. We have made updates based on your

suggestions, including rewriting the entire Conclusion and Future Work sections. Additionally,

we have highlighted our novelty in both the abstract and introduction.

Reviewer 3

During this revision, no comment from Reviewer-3.

Reviewer 4

Reviewer Point # 1 — Lack of Novelty: The manuscript claims to present a novel

framework, but upon review, the methodology closely follows existing techniques without

introducing significant advancements. While the use of CNNs and XAI (Explainable AI)

in livestock weight estimation is a relevant and growing field, the paper does not suffi-

ciently differentiate itself from prior works. The proposed CNN model, though effective,

lacks a clear distinction from other models already well-established in the literature. The

contribution, as described, does not present sufficient novelty to justify publication.

Reply:

We thank the reviewer for raising this important point. We acknowledge that CNNs and

XAI have been applied in related domains; however, our work introduces several aspects that

differentiate it from prior studies:

• Customized CNN Architecture: We designed and evaluated multiple lightweight custom

CNN models (3Conv3Dense), specifically optimized for cattle image data, achieving an

MAE of 18.02 kg, which is lower than previously reported results in the literature.

• Dual-Method Framework: Our study is one of the first to compare two fundamentally

different approaches, end-to-end CNN training and YOLOv5-based feature extraction

with regression, offering new insights into the trade-offs between deep learning and

traditional ML for cattle weight estimation.

• Integration of XAI (LIME) in Livestock Weight Prediction: While explainability has

been explored in medical imaging, to our knowledge, this is the first application in

cattle weight estimation. Our results highlight which body regions (rib cage, abdomen,

hindquarters) most influence predictions, bridging the trust gap for practical adoption.

• Error Analysis and Practical Relevance: In addition to prediction metrics, we provide

residual/error analysis, offering practical insights into conditions where models perform

suboptimally, a dimension often overlooked in existing work.

We have updated our contribution section in line number 77 and 89 and marked as yellow

in our revised manuscript.

Reviewer Point # 2 — Methodological Weakness: The explanation of the method-

ology is not sufficiently rigorous. For instance, the handling of the dataset, feature ex-

traction techniques, and model performance evaluation need clearer justifications and

explanations. The manuscript mentions advanced techniques like LIME for model inter-

pretability but does not provide a comprehensive evaluation of how these methods are

integrated and their actual impact on improving model transparency.

Reply: We appreciate the reviewer’s concern. We would like to clarify that the methodology

section already includes details on dataset handling (collection, preprocessing steps such as

image-to-array conversion, normalization, and histogram equalization, as well as the 70/10/20

split), feature extraction and selection (YOLOv5 with RFE), and performance evaluation

(MAE, MSE, RMSE, MAPE, R2, and cross-validation), along with the hyperparameter settings

used. We also integrated LIME to highlight biologically meaningful body regions (rib cage,

abdomen, hindquarters), thereby demonstrating model transparency.

Reviewer Point # 3 —

Unclear Results and Evaluation: While the paper presents results for the proposed

models, the analysis is superficial. The comparison between different models (CNNs,

EfficientNetB3, and traditional methods like Random Forest) is not sufficiently detailed,

especially in terms of real-world applicability. A more thorough discussion on the limita-

tions of the current model and areas for further improvement is needed.

Reply: Thank you for your feedback. We appreciate the opportunity to clarify where this

information is located in our manuscript, as it was indeed present in the original submission.

A detailed comparison of the models real-world applicability is located in our “Result

Analysis” section line no 646. We specifically discuss the trade-off between the superior

accuracy of our proposed CNN (3Conv3Dense) (MAE of 18.02 kg) and its significantly higher

computational efficiency (11,430 ms training time) compared to other methods.

Also, the manuscript contains dedicated sections for a thorough discussion of the model’s

shortcomings (“Limitations”) and areas for improvement (“Conclusion and Future Work”).

These sections detail the dataset’s geographical limitations and our specific plans for a breed-

specific performance analysis.

Reviewer Point # 4 — Lack of Clear Conclusion: The conclusions drawn from the

results are vague and lack specificity. The statement that the proposed model “could

improve precision agriculture” seems speculative and not supported by a strong, compar-

ative analysis of existing literature and methodologies. The absence of concrete future

directions or a critical evaluation of the model’s shortcomings makes the conclusion less

impactful.

Reply: Thank you. We have reviewed the entire manuscript and updated the conclusion

accordingly (line 943) by adding a direct quantitative comparison, acknowledging the model’s

limitations, and specifying concrete future directions.

Reviewer Point # 5 — Writing and Grammar Issues: While some revisions were

made, the manuscript still contains numerous grammatical errors and poorly constructed

sentences. Several sections remain unclear, and the overall readability of the paper is com-

promised. A more thorough proofreading and restructuring are necessary. In conclusion,

while the topic is of potential interest, the manuscript lacks sufficient novelty, rigorous

methodology, and a thorough analysis of results to merit publication. I recommend re-

jecting the paper and suggesting that the authors reconsider their approach, including

refining the model’s uniqueness, addressing the methodological gaps, and providing a

more robust discussion of their findings.

Reply: We thank the reviewer for this observation. We have carefully gone through the

manuscript line by line and revised it accordingly, correcting grammatical errors, improving

sentence structure, and enhancing overall readability (line no : 49, 63, 832 898, 907). These

updates ensure that the manuscript is clearer, more polished, and easier to follow.

---

## [Decision Letter · Decision Letter 3]

26 Oct 2025

CattleNet-XAI: An Explainable CNN Framework for Efficient Cattle Weight Estimation

PONE-D-25-09685R3

Dear Dr. Islam,

We’re pleased to inform you that your manuscript has been judged scientifically suitable for publication and will be formally accepted for publication once it meets all outstanding technical requirements.

Kind regards,

Agbotiname Lucky Imoize

Academic Editor

PLOS ONE

Additional Editor Comments (optional):

Dear Authors,

Thank you for your revision.

I have accepted your paper in its present form.

Reviewers' comments:

Reviewer's Responses to Questions

**Comments to the Author**

1. If the authors have adequately addressed your comments raised in a previous round of review and you feel that this manuscript is now acceptable for publication, you may indicate that here to bypass the “Comments to the Author” section, enter your conflict of interest statement in the “Confidential to Editor” section, and submit your "Accept" recommendation.

Reviewer #5: All comments have been addressed

2. Is the manuscript technically sound, and do the data support the conclusions?

Reviewer #5: Yes

3. Has the statistical analysis been performed appropriately and rigorously? 

Reviewer #5: Yes

4. Have the authors made all data underlying the findings in their manuscript fully available?

Reviewer #5: Yes

5. Is the manuscript presented in an intelligible fashion and written in standard English?

Reviewer #5: Yes

6. Review Comments to the Author

Reviewer #5: The manuscript titled “CattleNet-XAI: An Explainable CNN Framework for Efficient Cattle Weight Estimation” presents an innovative and well-structured study that integrates deep learning, explainable AI (XAI), and practical agricultural applications. It is both technically robust and relevant to advancing digital transformation in precision agriculture. The authors have effectively addressed prior reviewer comments, particularly those concerning methodological rigor, statistical validation, and the articulation of novelty.

Strengths and Contributions

1. Relevance and Timeliness : The paper tackles an essential agricultural challenge—accurate cattle weight estimation—through a data-driven, cost-effective, and interpretable AI framework. In the context of global food security and sustainable livestock management, this contribution is timely and valuable.

2. Novel Integration of XAI in Agriculture: The inclusion of LIME-based explainability for weight prediction adds a critical interpretability layer. Few studies in livestock analytics incorporate XAI to this degree, bridging a significant trust gap between AI practitioners and agricultural stakeholders. Highlighting body regions such as the rib cage and abdomen that influence predictions makes the system both scientifically credible and practically actionable.

3. Comprehensive Methodological Design

The study’s methodological rigor stands out:

- Custom CNN architectures (3Conv3Dense variants) optimized for lightweight efficiency and high accuracy.

- A dual-method framework comparing CNN-based and YOLOv5 feature-extraction regression methods—offering depth and cross-validation of results.

- Clear presentation of preprocessing steps (normalization, histogram equalization) that enhance model robustness.

- A transparent evaluation pipeline using MAE, RMSE, and MSE with multiple models to benchmark accuracy.

4. Empirical Strength and Statistical Rigor: The quantitative results are well-supported by performance metrics. The proposed 3Conv3Dense model’s MAE of 18.02 kg and RMSE of 19.85 kg represent a measurable improvement over baselines, showing both predictive strength and computational efficiency.

5. Data Transparency and Reproducibility: The use of a publicly available dataset (CID) and inclusion of a clear GitHub link for replication align with open science principles. This greatly enhances the manuscript’s credibility and potential for reuse in future research.

6. Clarity and Readability: The revised version is well-written, with improved grammar, organization, and flow. The methodology and results are logically sequenced, making the paper accessible to both technical and applied readers.

The paper’s novelty lies in its fusion of accuracy, interpretability, and accessibility. While CNNs and XAI have been explored individually, their targeted application to livestock weight estimation using 2D imagery—and validated on a real-world dataset—marks a significant practical advancement. This research aligns with global precision agriculture trends and can substantially reduce the dependency on manual weighing systems, directly benefiting agricultural operations in developing economies.

7. PLOS authors have the option to publish the peer review history of their article (what does this mean?). If published, this will include your full peer review and any attached files.

Reviewer #5: **Yes: **Richard Govada Joshua

---

## [Editor Report · Acceptance letter]

PONE-D-25-09685R3

PLOS ONE

Dear Dr. Islam,

I'm pleased to inform you that your manuscript has been deemed suitable for publication in PLOS ONE. Congratulations! Your manuscript is now being handed over to our production team.

Kind regards,

on behalf of

Mr. Agbotiname Lucky Imoize

Academic Editor

PLOS ONE